# Consistent histories of anthropogenic Western European air pollution preserved in different Alpine ice cores

Anja Eichler[1,2], Michel Legrand[3,4], Theo M. Jenk[1,2], Susanne Preunkert[3], Camilla Andersson[5], Sabine Eckhardt[6], Magnuz Engardt[7], Andreas Plach[8], Margit Schwikowski[1,2,9]

[1] Laboratory of Environmental Chemistry, Paul Scherrer Institute, CH-5232 Villigen PSI, Switzerland
[2] Oeschger Centre for Climate Change Research, University of Bern, CH-3012 Bern, Switzerland
[3] Institut des Géosciences de l'Environnement, Université Grenoble Alpes, Grenoble, France
[4] Laboratoire Interuniversitaire des Systèmes Atmosphériques, Université de Paris and Univ Paris Est Creteil, CNRS, LISA, F-75013, France
[5] Swedish Meteorological and Hydrological Institute, SE-60176 Norrköping, Sweden
[6] NILU - Norwegian Institute for Air Research, Kjeller, Norway
[7] Environment and Health Administration, SE-10420 Stockholm, Sweden
[8] Department of Meteorology and Geophysics, University of Vienna, Austria
[9] Department of Chemistry and Biochemistry, University of Bern, CH-3012 Bern, Switzerland

*Correspondence to*: Anja Eichler (anja.eichler@psi.ch)

**Abstract.** Individual high-Alpine ice cores have been proven to contain a well preserved history of past anthropogenic air pollution in Western Europe. The question, how representative one ice core is with respect to the reconstruction of atmospheric composition in the source region, has not been addressed so far. Here, we present the first study systematically comparing longer-term ice-core records (AD 1750-2015) of various anthropogenic compounds, such as major inorganic aerosol constituents ($NH_4^+$, $NO_3^-$, $SO_4^{2-}$), black carbon (BC), and trace species (Cd, $F^-$, Pb). Depending on the data availability for the different air pollutants, up to five ice cores from four high-Alpine sites located in the European Alps analysed by different laboratories were considered. Whereas absolute concentration levels can partly differ depending on the prevailing seasonal distribution of accumulated precipitation, all seven investigated anthropogenic compounds are in excellent agreement between the various sites for their respective, species-dependent longer-term concentration trends. This is related to common source regions of air pollution impacting the less than 100 km distant four sites including Western European countries surrounding the Alps. For individual compounds, the Alpine ice-core composites developed in this study allowed us to precisely time the onset of pollution caused by industrialization in Western Europe. Extensive emissions from coal combustion and agriculture lead to an exceeding of pre-industrial (AD 1750-1850) concentration levels already at the end of the 19[th] century for BC, Pb, $exSO_4^{2-}$ (non-dust, non-sea salt $SO_4^{2-}$) and $NH_4^+$, respectively. However, Cd, $F^-$, and $NO_3^-$ concentrations started surpassing pre-industrial values only in the 20[th] century, predominantly due to pollution from zinc and aluminium smelters and traffic. The observed maxima of BC, Cd, $F^-$, Pb, and $exSO_4^{2-}$ concentrations in the 20[th] century and a significant decline afterwards, clearly reveal the efficiency of air pollution control measures such as desulphurisation of coal, the introduction of filters and scrubbers in power plants and metal smelters, and the ban of leaded gasoline improving the air quality in Western Europe. In

contrast, $NO_3^-$ and $NH_4^+$ concentration records show levels in the beginning of the $21^{th}$ century, which are unprecedented in the context of the past 250 years, indicating that the introduced abatement measures to reduce these pollutants were insufficient to have a major effect at high altitudes in Western Europe. Only four ice-core composite records (BC, $F^-$, Pb, $exSO_4^{2-}$) of the seven investigated pollutants correspond well with modelled trends, suggesting inaccuracies of the emission estimates or an incomplete representation of chemical reaction mechanisms in the models for the other pollutants. Our results demonstrate that individual ice-core records from different sites in the European Alps generally provide a spatial representative signal of anthropogenic air pollution trends in Western European countries.

## 1 Introduction

High-alpine glaciers and polar ice sheets are invaluable archives of past air pollution and climate. Since direct measurements of the atmospheric composition are basically not available before the 1970s (WMO GAW, 2014), these natural archives are unique to put recent observations into a longer-term context and assess the anthropogenic impact in relation to natural changes in atmospheric constituents.

Ice-core records extending back to pre-industrial times were extracted from glaciers around the world, illustrating that various regions experienced distinct timing of anthropogenic emissions over the past ~150 years reflecting significant differences in industrialization and emission abatement. European, Greenland, and Devon Island ice-core records have been used to reconstruct the history of anthropogenic pollutants such as $NH_4^+$, $NO_3^-$, Pb, and $SO_4^{2-}$ in Western Europe and North America, characterized by a substantial increase beginning already in the mid-$19^{th}$ century (Fischer et al., 1998; Schwikowski, 2004; Shotyk et al., 2005). Air pollutant levels above natural background in Eastern Europe appeared around 1930 based on ice-core records from the Altai and Caucasus, indicating a later onset of industrialization (Olivier et al., 2006; Eichler et al., 2014; Preunkert et al., 2019a). Concentrations of the majority of anthropogenic air pollutants in Western Europe, North America, and Eastern Europe peaked in the second half of the $20^{th}$ century (Fischer et al., 1998; Schwikowski, 2004; Olivier et al., 2006). The subsequent downward trend is the consequence of changes in fuel type used (from coal to oil and gas), introduction of abatement measures, but also stagnation and downturn in industry after the political, social, and economic instability in the case of Eastern Europe (Eichler et al., 2012). On the contrary, anthropogenic emissions in Asia as reconstructed from Himalayan, Pamir, and Saint Elias Mountain ice-core records increased only after AD 1960-1970 and are still rising at the beginning of the $21^{th}$ century, due to a later industrialisation in these regions (Duan et al., 2007; Osterberg et al., 2008; Zhao et al., 2011).

Ice-core drilling and analyses are very costly, demanding and time consuming processes, taking from months up to years. Thus, generally only one ice core is recovered at a time and replicates are rarely taken. It is assumed that one core is representative for emissions in a particular source region. However, there is a lack of evidence in literature, for which ice-core sites this assumption is actually valid. Only for Greenland ice cores a strong covariance in the trends of black carbon (BC), $NO_3^-$, Pb, and $SO_4^{2-}$ records was observed between close-by sites (less than 100 km) and also for locations being 1000 km

apart (Fischer et al., 1998; Koch et al., 2011; Zdanowicz et al., 2018; Mcconnell et al., 2019). This finding indicates common source regions for the different Greenland sites, located in North America and Europe, characterised by a similar temporal evolution of anthropogenic emissions. In contrast, for other Arctic sites, such as the Saint Elias Mountains or Svalbard, significant differences in the anthropogenic trends of BC, $NO_3^-$, Pb, or $SO_4^{2-}$ were detected among ice cores from close-by locations (Gross et al., 2012; Samyn et al., 2012; Beaudon et al., 2013; Osmont et al., 2018). Within the Saint Elias Mountains a distinct vertical profile of aerosol deposition was found. Ice cores from 5000 m above sea level (asl) documented long-range transported air pollution from East Asia, whereas those from 3000 m asl additionally recorded local emissions from North America (Gross et al., 2012). In Svalbard a west-east pollution gradient was detected, with ice cores from western parts reflecting rather boundary layer pollution, whereas those from eastern glaciers are more representative of free tropospheric conditions (Beaudon et al., 2013). Additionally, the latter study reports strong post-depositional melt-water induced disturbance of the ice-core records at the low-altitude Svalbard sites. Differences in $SO_4^{2-}$, Pb or Sb concentration trends have also been observed from two Himalayan ice cores (Mount Everest, Dasuopu, ~100 km distant) (Duan et al., 2007; Kaspari et al., 2009; Gabrielli et al., 2020). Although pollutants from South Asian regions are deposited at both sites, the ~900 m higher elevated Dasuopu probably receives a larger share of long-range transported air masses. Furthermore, in the Himalayas overwhelming natural dust deposition leads to an additional input of the compounds often strongly exceeding their anthropogenic impact (Sierra-Hernandez et al., 2018). For such locations an extensive dust correction is required, potentially producing higher uncertainties in the calculated anthropogenic trends. In conclusion, for many mountain ranges or glacier sites one ice core is not representative with respect to the reconstruction of atmospheric composition in a given source region.

This setting is different for the European Alps. Here, high-Alpine ice-core sites are within a short distance of less than 100 km, altitude differences are <500 m, and the signal of anthropogenic emissions in the ice-core records is overwhelming during industrial times. Individual ice cores from different high-altitude sites in the European Alps have been proven to contain well-preserved records of past Western European air pollution (e.g. (Schwikowski et al., 1999a; Schwikowski et al., 1999b; Eichler et al., 2000a; Fagerli et al., 2007; Preunkert and Legrand, 2013; Preunkert et al., 2019b). Generally, ice-core concentrations of anthropogenic major aerosol components and trace constituents increased dramatically during the last two centuries following the onset of industrialization (Schwikowski, 2004). Since the second half of the 20th century, concentrations for most of these species significantly declined as a result of the introduction of air pollution control measures in Western Europe. Two comparative studies indicate a good correspondence between $NO_3^-$ and $SO_4^{2-}$ records of summer concentrations at the Col Du Dôme site and annual concentrations from the 80 km distant Colle Gnifetti site in the period ~AD 1910-1995 (Preunkert et al., 2000; Preunkert et al., 2003; Wagenbach et al., 2012) and between the concentration trends of BC records from the Colle Gnifetti and 70 km distant Fiescherhorn site in the period ~AD 1750-2000 (Sigl et al., 2018), respectively. However, a systematic and more holistic investigation of pollution records from different ice cores to evaluate the spatial representativeness of single sites in the European Alps is still lacking.

Here we present the first study comparing longer-term records of anthropogenic pollution recovered from various European high-Alpine ice-core sites for the period AD 1750-2015. We focussed on pollutants, for which continuous records from at least

two of the four sites Colle Gnifetti, Fiescherhorn, Grenzgletscher (Swiss Alps) and Col du Dôme (French Alps) were available, i.e. the major inorganic aerosol species $NH_4^+$, $NO_3^-$, $exSO_4^{2-}$; BC, and the trace species Cd, $F^-$, Pb. This set of air pollutants analysed by different laboratories covers compounds of various emission characteristics and atmospheric chemistry. Among the seven considered pollutants, BC is the only one directly emitted to the atmosphere as primary aerosol from incomplete combustion processes. $NH_4^+$, $NO_3^-$, $SO_4^{2-}$ are the dominant species and Cd, $F^-$, Pb minor species in secondary inorganic aerosols. These are formed in the atmosphere by chemical reactions involving a set of precursor compounds and condensation of vapours on pre-existing particles or by nucleation of new particles. $NH_3$, NOx, $SO_2$ emissions from agriculture and burning of fossil fuels associated with transportation and industry represent the main anthropogenic precursor compounds for the major inorganic aerosol species (Seinfeld and Pandis, 2016). Emissions of Cd(0) or CdO, HF, and PbBrCl from zinc and aluminium smelters and leaded gasoline, respectively, are predominant precursors for the anthropogenic trace species (Biggins and Harrison, 1979; Williams and Harrison, 1984; Abdul-Wahab and Alsubhi, 2019). Chemical reactions of these precursors in the atmosphere include oxidations, such as the formation of $HNO_3$, $H_2SO_4$, Cd(II) compounds from NOx, $SO_2$, Cd(0), respectively. Emitted precursors or atmospheric reaction products either directly form particles (such as $H_2SO_4$) or partition into particles by e.g. acid-base reactions ($HNO_3$, $NH_3$, HF) or by different reaction types (for Cd, Pb compounds) (see e.g. (Biggins and Harrison, 1979)). After an atmospheric lifetime in the order of a few days (Seinfeld and Pandis, 2016), wet (and dry) deposition lead to their removal from the atmosphere.

Ice-core records of the seven pollutants were systematically compared in terms of their concentration levels and the observed longer-term changes, to test how representative a single ice core from the European Alps documents the pollution history in the source region. The pollution source regions for the four Alpine sites were determined using the state-of-the-art atmospheric aerosol transport model (FLEXPART). For the investigation of differences in absolute concentration levels at the four ice-core sites, the European-scale atmospheric chemical transport model (MATCH) was used to simulate concentrations of $NH_4^+$, $NO_3^-$ and $exSO_4^{2-}$ in Alpine precipitation. In addition, we present Alpine composite records for the seven species and their evaluation using time of emergence (TOE) analyses, revealing the time at which Western European pollution levels first exceeded the natural variability of the pre-industrial period (AD 1750-1850) and to place recent changes into a longer-term perspective. Finally, the ice-core composites of all pollutants were compared with available model estimates of either their emission history, atmospheric concentrations, or deposition data allowing to validate and constrain existing approaches for modelling the development of anthropogenic air pollution in Western Europe.

## 2 Methods

### 2.1 Study sites and ice-core archives

Longer-term ice-core data are available from four sites located in the Swiss and French Alps: Monte Rosa in the Valais Alps (Colle Gnifetti and Grenzgletscher), Fiescherhorn in the Bernese Alps (Fiescherhorn glacier) and Mont Blanc in the Auvergne-

Rhône-Alpes (Col du Dôme). These four Alpine sites are located within a ~100 by 100 km square. Their exact locations are shown in Figure 1 with details about the different ice cores being summarized in Table 1.

### 2.1.1 Colle Gnifetti (CG)

The Colle Gnifetti (4450-4470 m asl) is located in the Monte Rosa massif on the border between Switzerland and Italy in the Southern Alpine chain (Valais Alps). The Colle Gnifetti is a glacier saddle at the uppermost part of the Grenzgletscher between the summits Signalkuppe and Zumsteinspitze. Numerous firn and ice cores have been drilled there since 1976, revealing that this site is prone to wind erosion, preferentially removing parts or all of the winter snow cover (Wagenbach et al., 2012). The latter explains the low mean annual net accumulation rates of 0.2-0.5 m water equivalent (w.eq.) (Table 1) and the observation that older than 15000-year old ice was found at the bedrock (Jenk et al., 2009; Hoffmann et al., 2018). The dating of the different cores used in this study was achieved by combining a variety of dating methods such as annual layer counting using the seasonally varying signals of different parameters (e.g. $NH_4^+$ and $\delta^{18}O$), $^{210}Pb$ dating, the use of reference horizons (Saharan dust events, 1963 $^3H$ maximum from atmospheric nuclear tests, volcanic eruptions), application of ice flow models, and $^{14}C$ dating (Jenk et al., 2009; Sigl et al., 2018). In this study, we focus on the ice-core data covering the period AD 1750-2015, as detailed in Table 2.

### 2.1.2 Grenzgletscher (GG)

The Grenzgletscher was the main tributary of the Gornergletscher system in the Monte Rosa massif, until the strong retreat of the Gornergletscher separated them recently. A 125 m long ice core was drilled at 4200 m asl on the upper Grenzgletscher in October 1994 (Eichler et al., 2000b). The ice core covers the period AD 1937-1994 with a mean annual net accumulation rate of 2.7 m w.eq. (Table 1). This was established based on $^{210}Pb$ dating, annual layer counting, and detection of reference horizons, such as the Saharan dust layers in AD 1947, 1977, and 1990, as well as the $^{137}Cs$ maxima in AD 1986 and 1963 from the Chernobyl accident and atmospheric nuclear testing, respectively (Eichler et al., 2000b). Records of chemical species are partly disturbed for the period AD 1984-89, due to an inflow and percolation of meltwater causing partial leaching (Eichler et al., 2001). Thus, for this study only the undisturbed section covering the period AD 1937-1983 was considered.

### 2.1.3 Fiescherhorn (FH)

The Fieschergletscher in the Swiss Northern Alpine chain (Bernese Alps) has an extended plateau flanked by the mountains Grosses Fiescherhorn, Hinteres Fiescherhorn, and Kleines Fiescherhorn. A 150.5 m long ice core was drilled in 2002 at an altitude of 3900 m asl (Jenk et al., 2006). Dating was performed by annual layer counting, time markers such as Saharan dust events in AD 1947, 1977, 2000 and the 1912 Katmai volcanic layer, and ice flow modelling. The ice core covers the period from AD 1660 to 2002 and the mean annual net accumulation rate is 1.4 m w.eq. (Jenk, 2006; Jenk et al., 2006). Similar to the Grenzgletscher site, concentration records of selected chemical species were partly disturbed between AD 1989 and 1997 (Jenk, 2006), and we therefore restricted discussions to the period AD 1750-1988.

### 2.1.4 Col du Dôme (CDD)

The Col du Dôme is a glacier saddle at 4250 m asl between the Mont Blanc main summit and the Dôme de Goûter in the French Alps. Data used in this study are from three ice cores drilled in 1994 (126 m), 2004 (124 m), and 2012 (140 m). Ice-core sites are less than 20 m apart. Dating was performed by annual layer counting mainly based on the $NH_4^+$ concentrations (Preunkert et al., 2000), refined with reference horizons, such as the Saharan dust layers in 1936/37, 1947, and 1977, the 1963 $^3H$ maximum, and the $^{137}Cs$ maxima in 1954 and 1986. The mean annual net accumulation rate near the surface is 2.4 m w.eq.,

with a regular distribution of seasonal accumulation (Vincent et al., 1997). Due to changes in depositional processes upstream of the drilling site, winter layers at the CDD site were found to generally thin with depth relative to summer layers (Preunkert et al., 2000), resulting in a winter to summer accumulation ratio of 1:1 at the surface and of 0.4:1 in 1925 and to an absence of winter snow layers prior to 1890 (Legrand et al., 2020). To account for this, in the present study only data for the period AD 1890-2012 were considered.

## 2.2 Sampling and analyses

### 2.2.1 $NH_4^+$, $NO_3^-$, $SO_4^{2-}$, and $F^-$

The determination of the major ion concentrations ($NH_4^+$, $NO_3^-$, $SO_4^{2-}$, $Ca^{2+}$, $Na^+$) and of $F^-$ in the CDD, CG15, CG93, FH, and GG cores on discrete samples was performed by Ion Chromatography (IC) with conductivity detection at the Paul Scherrer Institute (PSI) (Eichler et al., 2000a; Eichler et al., 2000b), the Institute for Environmental Physics, Heidelberg (IUP), and the

Laboratoire de Glaciologie et Geophysique de l'Environnement, Grenoble (LGGE) (Legrand et al., 1993; Preunkert et al., 2001a).

### 2.2.2 Pb and Cd

Pb and Cd concentrations were determined in discrete samples by Inductively Coupled Plasma Sector Field Mass Spectrometry (ICP-SFMS) at the PSI (CG95 core) (Schwikowski et al., 2004), by continuous flow analyses (CFA) with Inductively Coupled

Plasma Quadrupole Mass Spectrometry (ICP-QMS) at the University of Venice (CG03 core) (Gabrieli, 2008; Gabrieli and Barbante, 2014) and by CFA ICP-SFMS at the Desert Research Institute (DRI) (CDD cores) (Legrand et al., 2020). ICP-SFMS and Laser Ablation Inductively Coupled Plasma Mass Spectrometry (LA-ICP-MS) at the University of Maine was used for the analyses of the CG13 core (More et al., 2017).

### 2.2.3 Black carbon - rBC and EC

Concentrations of atmospheric soot particles are determined with different analytical techniques. Depending on the method used, the non-organic carbon fraction of soot is denoted as BC or elemental carbon (EC). We here followed the notation recommended by (Petzold et al., 2013). After aerosolizing and drying of the aqueous sample in a jet nebulizer (APEX-Q, Elemental Scientific Inc., USA), concentrations of BC in the CG15 core were determined based on laser-induced

incandescence with a Single Particle Soot Photometer (SP2, Droplet Measurement Technologies) and are in the following
accordingly denoted as refractory black carbon (rBC)  (Wendl et al., 2014; Sigl et al., 2018). Concentrations in the FH core
were either measured by a thermal (Jenk et al., 2006) or thermal/optical technique (Cao et al., 2013) and thus here denoted as
EC. For samples covering the period AD 1660-1940, EC was combusted to $CO_2$ for subsequent manometric quantification
(Jenk et al., 2006), while it was quantified by a non-dispersive infrared (NDIR) detector (OC/EC analyzer, Sunset Laboratory
Inc., USA) for the more recent samples (Cao et al., 2013).

**2.3 Data presentation, formation of composite records, time of emergence analyses**

The datasets for the four individual Alpine ice-core sites were compiled from values reported previously in the studies listed
in Table 2. We here used data of the seven investigated species (BC, Cd, $F^-$, $NH_4^+$, $NO_3^-$, Pb, $SO_4^{2-}$) published before end of
2021. For $SO_4^{2-}$ the non-dust and non-sea salt contribution ($exSO_4^{2-}$) was calculated from the total $SO_4^{2-}$ accordingly to
$[exSO_4^{2-}] = [SO_4^{2-}]_{total} - 0.12[Na^+] - 0.175 [Ca^{2+}]$, using $Ca^{2+}$ and $Na^+$ as mineral dust and sea salt tracers, respectively
(Schwikowski et al., 1999b). $ExSO_4^{2-}$ thus represents the part of $SO_4^{2-}$, exclusively formed from the atmospheric oxidation of
$SO_2$.
For all ice cores, annual averages were calculated similarly from these non-equidistant data as arithmetic mean of all data
points in an annual layer. Then, based on these annual means, 10-year arithmetic concentration averages were calculated,
which are representative of pollutant emissions, smoothing out inter-annual (short-term) fluctuations related to temperature
dependent vertical transport to the high-alpine sites (see S1 and Fig. S1). Furthermore, the used 10-year means account for a
lower temporal sampling resolution and increasing dating uncertainty with increasing depth (e.g. 10-25 years for the FH in the
period 1750-1800). Averages represent mainly decadal means (e.g. 2000-2009), except for the most recent period (2010-2015).
Previous studies from CDD used a different procedure to calculate annual means. To account for the observed longer-term
trend in changing preservation of winter relative to summer snow (see section 2.1.4.), concentrations of winter and summer
layers were averaged (Legrand et al., 2020) or winter and summer records investigated separately (Preunkert et al., 2001a;
Preunkert et al., 2003; Preunkert and Legrand, 2013). Since for the other three study sites a disentangling of winter and summer
layers was not possible especially for the pre-industrial periods with low data resolution, we accordingly formed arithmetic
means of total annual layer data for all ice cores consistently as described above. In any case, for the 10-year means considered
in this study, CDD longer-term concentration trends for both averaging procedures were found to agree within the uncertainty
envelopes for all investigated species (see S2 and Fig. S2).
Based on the 10-year average records obtained for each site, an ice-core composite record was formed for each species. For
this, individual records were transformed into z-scores using the average and standard deviation of the common period AD
1910-1990. To obtain a sufficiently long overlapping period, the relatively short (< 50 years) GG records were not included.
The BC composite record consists of the FH EC and CG rBC records.
TOE analyses was applied to determine when the composite z-scores exceed the natural variability of the pre-industrial period
for the first time. The TOE was defined as the year when the 10-yr-average composite z-scores first exceed (and remain above)

the pre-industrial average z-scores (period AD 1750-1850) by at least two standard deviations determined from the variability in the pre-industrial 10-year averages (Lehner et al., 2017).

## 2.4 Source region of air pollutants using FLEXPART

The potential source regions of the pollutants deposited at the CDD, CG/GG, and FH sites were derived using the state-of-the-art Lagrangian particle dispersion model FLEXPART (version 10.4) (Stohl et al., 2005; Pisso et al., 2019). The model maps the emission sensitivity of sub-micrometre aerosol particles deposited at the three sites (Figure 1). The emission sensitivity in a certain grid cell represents the ratio of aerosol deposition at the respective study site (in $\mu g\ m^{-2}\ a^{-1}$) to emission of this aerosol in the grid cell (in $kg\ s^{-1}$) ("receptor/source" ratio). A detailed description of the calculation of the source-receptor relationship

for deposited mass in FLEXPART is provided by Eckhardt et al. (2017). In the atmosphere over the Alps all species discussed in this study are predominantly present as sub-micrometre aerosols. The model was therefore run for $SO_4^{2-}$ as a target species for dispersion of sub-micrometre aerosol. FLEXPART was run in its backward-in-time mode simulating dry and wet deposition (Eckhardt et al., 2017) for the period 2000 through 2009, while employing ERA5 re-analysis data at 0.5° x 0.5° horizontal and 1-hourly temporal resolution (137 vertical layers) (Hersbach et al., 2018) as the meteorological input. Particles

were released continuously ($10^6/10^5$ per month for wet/dry deposition, respectively) and traced backward for 30 days. FLEXPART uses a statistic approach to calculate the mean residence time of all released particles in individual grid cells to derive the emission sensitivity. In order to compensate for model surface misrepresentation (the Alps are lower in the model than reality), wet deposition, which is highly sensitive to altitude, was calculated at the actual altitudes of the sites.

## 2.5 Modelling of air pollutant concentrations in precipitation

Two different versions of a European-scale atmospheric chemical transport model, MATCH (Robertson et al., 1999; Andersson et al., 2007; Theobald et al., 2019) were used to simulate concentrations of $NH_4^+$, $NO_3^-$ and $exSO_4^{2-}$ in precipitation at the ice-core sites. To construct secular concentration trends in the period 1900-2020, we utilised a well-described set-up (50 km resolution) developed for the ECLAIRE project (MATCH-ECLAIRE). Details on this simulation, including input data, can be found in Engardt et al. (2017). The spatial distribution of $exSO_4^{2-}$ concentration during the 10-year periods 1970-1979

and 1990-1999 in the Alps were investigated using MATCH-ECLAIRE and for the more recent decade 1990-1999 also with the new, high-resolution set-up MATCH-BIODIV. The MATCH-BIODIV domain extends over most of Europe at 12 km resolution and the dataset covers the period 1987-2020. MATCH-BIODIV was forced by meteorological data generated by the regional climate model ALADIN with EC-EARTH on its boundaries under the influence of greenhouse gas emission scenario RCP8.5 (Belušić et al., 2020; Lind et al., 2022), and nitrogen and sulphur emissions from ECLIPSE V6b (Höglund-

Isaksson et al., 2020). The output from the chemical transport model was finally averaged over the same 10-year periods as the ice-core data.

## 3 Results and Discussion

Average emission sensitivities during the period 2000-2009 derived with FLEXPART, indicate that in general the four ice-core sites CDD, CG, FH, and GG are most sensitive to emissions from the countries surrounding the Alps, i.e. Switzerland,

France, Italy, Germany, Austria, Slovenia, and Spain, without substantial discrepancies between the four sites (Figure 1). A noticeable extension of the pollution source area into Spain is recognized for the westernmost CDD, and into larger parts of Germany for the northernmost FH site. To investigate, how representative ice-core records from the four sites are to document the pollution history in the identified source area we systematically compared a) concentration levels (section 3.1.) and b) longer-term variations in concentrations of BC, Cd, $F^-$, $NH_4^+$, $NO_3^-$, Pb, and $exSO_4^{2-}$ (section 3.2.).

### 3.1 Concentration levels

Absolute concentrations are very similar at CG, FH, and GG, whereas CDD concentrations are about a factor of 1.4 lower on average (1.1-1.7, Figure 2). Potential reasons for the lower concentrations at CDD are (1) a west-east gradient in the deposition of anthropogenic pollutants in the Alps, and/or (2) a relative higher amount of the cleaner winter snow at the CDD site.

(1) Increasing concentrations of $NH_4^+$, $NO_3^-$, and $SO_4^{2-}$ were observed along an Alpine west-east transect in winter snow

samples for the period AD 1991-1993 (Nickus et al., 1997). The findings of this short-term study are supported by MATCH simulations using the MATCH-ECLAIRE set-up (50 km resolution, available for the period AD 1900-2020) which produced a factor of 2-3 difference between $exSO_4^{2-}$ concentration in precipitation at CDD and the other sites (Figure 3, Table 3). However, simulated concentrations are a factor of 6-8 higher than the ice-core values. Due to the rather low spatial resolution in MATCH-ECLAIRE, the mean heights of the grid cells including the drilling sites are between 1600 and 1800 m asl, far

lower than the actual height of the sites (3900-4450 m asl). Generally, pollution concentrations in precipitation are higher at lower altitudes, where the emission sources are located. To test, whether the model output depends on the spatial resolution of the chemical transport model and its input (meteorology, emissions, topography etc.), we used data from MATCH-BIODIV with the higher spatial resolution of 12 km, providing concentrations in precipitation from 1987 on. Here, the mean height of the grid cells representing the ice-core sites span 2400 to 2700 m asl. As expected, $exSO_4^{2-}$ concentrations obtained from the

MATCH-BIODIV dataset for the period AD 1990-1999 are significantly lower compared to MATCH-ECLAIRE, and are comparable to the ice-core concentrations (Figure 3, Table 3). Furthermore, the simulated concentrations using MATCH-BIODIV are similar at the four sites and the high-Alpine region in general (green region in Figure 3e) supporting the FLEXPART results that the same source regions of pollutants impact these high-Alpine sites (see 3.1.1.). Thus, based on the MATCH-BIODIV results we can exclude a high-Alpine west-east concentration gradient of anthropogenic pollutants to

explain the lower concentrations at CDD.

(2) If not related to a west-east gradient, the lower concentrations observed at CDD could be explained by different conditions affecting snow deposition. In general, concentrations of anthropogenic pollutants at high elevation sites reveal a pronounced seasonal cycle due to a seasonality in the stability of the atmosphere. In summer, more intense radiation causes increased

convection and instability of the lower atmosphere, resulting in an expansion of the planetary boundary layer to the high elevations. During the colder months, mountain sites become decoupled from the boundary layer in the lower-altitude, stronger polluted areas, and encounter free tropospheric, cleaner, conditions. As an example, mean $NH_4^+$, $NO_3^-$, and $SO_4^{2-}$ summer-to-winter concentration ratios of 14, 3.9, and 3.7, respectively, were observed at CDD during the period AD 1982-1991 (Preunkert et al., 2000). Thus, the mean annual concentrations at high-altitude sites strongly depend on the seasonality of accumulated precipitation. At the CDD site in the western Alps, the distribution of precipitation is rather regular throughout the year (Vincent et al., 1997). This is different at the FH site in the northern Alps, revealing a precipitation maximum in summer due to convection. At the GG in the southern Alpine chain precipitation from tropical-maritime air masses during foehn situations leads to maxima in spring and autumn (Eichler et al., 2004). The CG site is prone to wind erosion, preferentially removing parts or all of the dry and lighter winter snow cover (see section 2.1.1.). For these reasons, the hypothesis that the lower ice-core concentrations at the CDD compared to CG, FH and GG is related to a higher share of winter precipitation seems reasonable. The fact that mainly summer deposition is preserved at the CG is corroborated by the agreement in concentration levels of different major ions between summer CDD and annual CG accumulation layers (Preunkert et al., 2000; Preunkert et al., 2003; Wagenbach et al., 2012). Because of the preferential loss of the cleaner winter snow, higher annual average concentrations levels at CG are expected also compared to FH and GG. However, similar concentrations levels at all three sites suggest that the loss of the cleaner winter snow at CG is compensated by the altitude concentration gradient (lower concentrations with increasing altitude) (Preunkert et al., 2001a).

Unfortunately, local differences in the seasonal distribution of precipitation and small-scale processes such as wind erosion affecting its deposition on the glacier cannot be tested in the MATCH model due to its limitations in resolution.

## 3.2 Longer-term concentration variations

Generally, an excellent agreement in the 10-year ice-core longer-term concentration changes (trends) of all investigated pollutants is observed for the four sites CDD, CG, FH, and GG (Figure 2). This is confirmed by correlation analyses, revealing high correlation coefficients ($0.84 < r < 1.00$) between the respective pollutant records from the different ice cores (Figure 4). All correlations are significant at the 0.001 level. The observed trends are characteristic for the individual seven pollutants, reflecting differences in their respective emission sources, emission histories, the atmospheric chemistry of precursor species, and compound specific implementation and efficiency of emission abatement measures. This is detailed in the following sections (3.2.1-3.2.3), including discussion of how representative the Western European pollution history is captured.

### 3.2.1 Trends in major inorganic aerosol species (exSO$_4^{2-}$, NO$_3^-$, NH$_4^+$)

**Excess sulfate**

The five alpine exSO$_4^{2-}$ records (CDD, CG15, CG93, FH, GG) consistently show low pre-industrial background levels (AD 1750-1850), increasing concentrations from the second half of the 19$^{th}$ century until the 1970s, and declining values afterwards

(Figure 2). In the pre-industrial period a significant proportion of the $exSO_4^{2-}$ originated from wood combustion for domestic heating (Schwikowski et al., 1999b). During the 1880s the Alpine composite $exSO_4^{2-}$ record indicates levels exceeding for the first time the variability of the pre-industrial values (Figure 5), denoting the onset of substantial anthropogenic $SO_2$ emissions from coal burning in Western Europe. In the following ~50 years FH concentrations are higher compared to the other sites Figure 2), which could reflect a stronger influence of $SO_2$ emissions from Germany that increased faster than in other Western

European countries during that time (Hoesly et al., 2018). After ~1930 there is a consistent increasing trend at all sites, cumulating in maximum values in the 1970s. The majority of the anthropogenic $SO_2$ emissions during that time was from fossil fuel combustion in energy and transformation industries (stationary sources). Decreasing $exSO_4^{2-}$ concentrations from the 1980s on are the consequence of the implementation of air pollution control measures, such as the desulphurisation of coal, the use of low-sulphur fuels, and the introduction of filters and scrubbers in power plants.

The observed longer-term trend in the composite $exSO_4^{2-}$ record and modelled $exSO_4^{2-}$ concentrations in precipitation at the drilling sites agree exceptionally well during the period AD 1900-2020 (Figures 3 and 5). This finding confirms that European $SO_2$ emissions and the $SO_2/SO_4^{2-}$ atmospheric chemistry are well described in the MATCH model. The pronounced maximum in the 1970s and the strong decline afterwards is reproduced on a regional, Western European level within the spatial model results (Figures 3b and 3c). The former is in accordance with the $SO_2$ emission peak in the main source regions (France,

Germany, Italy, and Switzerland) during the 1970s (Hoesly et al., 2018). The Alpine ice-core composite record and model estimates of $exSO_4^{2-}$ consistently demonstrate that concentrations in the 2010s were again as low as in the first half of the 20th century, confirming the high efficiency of the introduced abatement measures in Western Europe.

**Nitrate**

Five $NO_3^-$ records from CDD, CG15, CG93, FH, and GG are in excellent agreement, generally showing low pre-industrial
background levels from 1750 to the mid of the 20th century, rising values until the 1980s and stabilization during the past decades (Figure 2). Major emission sources during the pre-industrial period included NOx emissions from natural soils, lightning, wild fires and anthropogenic activities such as biofuel combustion and agricultural waste burning (Preunkert et al., 2003; Hoesly et al., 2018). The strong $NO_3^-$ increase in the second half of the 20th century reflects anthropogenic NOx emissions from high temperature combustion, pre-dominantly from the traffic and energy sector. The Alpine composite $NO_3^-$ record

exceeds the variability of the pre-industrial period for the first time during the 1930s (Figure 5), marking the onset of substantial anthropogenic NOx emissions to the atmosphere predominantly from motor vehicles in Western Europe. The delay of 50 years between TOE for $exSO_4^{2-}$ and $NO_3^-$ potentially reflects the change of fossil fuel use over time in Western Europe; from coal burning during the first half of the 20th century to liquid/gaseous fuel burning during the second half of the 20th century. Although air pollution control measures such as combustion catalysts have been widely implemented in recent decades in all

Western European countries, no prominent decline in the ice-core $NO_3^-$ concentrations is observed for the most recent years. During most of the 20th century, modelled concentrations in precipitation at the ice-core sites closely resemble the ice-core data (Figure 5). However, starting in the 1980s the composite $NO_3^-$ record diverges from the model results and estimated NOx

emissions from Western European countries (Hoesly et al., 2018). All relevant European countries show a significant drop of NOx emission estimates by roughly a factor of 2 between the 1980s and 2010s (Hoesly et al., 2018). Hypotheses for the disagreement are potential uncertainties in NOx emissions estimates or model misrepresentation of the $HNO_3/NO_3^-$ partitioning between gas and particle phase and of the atmospheric chemistry. E.g. the formation of nitrate aerosols from $HNO_3$ is depending on the presence of alkaline compounds (such as $NH_3$, or mineral dust). Still high recent $NH_4^+$ ($NH_3$) concentration levels (though not predicted by the MATCH model, see below) could favor the formation of $NH_4NO_3$ aerosols. Nevertheless, the still high Alpine composite $NO_3^-$ concentrations beginning of the 21[th] century indicate that the so far introduced air pollution control actions in Western Europe were still not sufficient to have a major effect on recent $NO_3^-$ levels at high altitudes.

**Ammonium**

Overall the available five $NH_4^+$ records from CDD, CG15, CG93, FH, and GG agree well, consistently indicating low pre-industrial values in the period AD 1750-1850, with a continuously increasing trend afterwards (Figure 2). The last ~30 years of the records are characterized by the highest $NH_4^+$ concentrations, unprecedented during the last 250 years. The major emission sources during pre-industrial times were biogenic and agricultural $NH_3$ emissions (Eichler et al., 2009). The observed differences between the CG15, CG93, and FH concentration records during the first 150 years are not related to differences in source regions, since the two CG cores are from the same site (Figure 2). This is most likely related to the fact that $NH_4^+$ measurements are prone to contamination from laboratory air (Legrand et al., 1984), which is especially problematic for the very low pre-industrial levels being often below 30 ppb. Increasing $NH_4^+$ levels starting from the second half of the 19[th] century were caused by rising anthropogenic $NH_3$ emissions from agriculture (bacterial decomposition of livestock wastes and fertiliser applications) (Döscher et al., 1996; Hoesly et al., 2018). Based on the TOE analysis of the Alpine composite, the onset of these extensive emissions in Western Europe started in the 1880s (Figure 5). Despite the partial introduction of agricultural emission reduction measures in Western Europe since the end of 20[th]/beginning of the 21[th] century, such as $NH_3$ reductions in fertilizer industry and livestock breeding (Giannakis et al., 2019; Liu et al., 2020; Liu et al., 2022), no corresponding decline in recent $NH_4^+$ levels is observed in the ice-core records. Indeed, $NH_3$ emission reductions targeted for the past ~20 years based on environmental policies were not achieved for many European countries (Giannakis et al., 2019). Observed slight decreasing $NH_3$ emissions in some countries during this time could be in part related to different factors than air pollution control such as the strong focus on animal welfare leading to declining livestock numbers (Kupper et al., 2015).

The observed longer-term trends in the Alpine composite $NH_4^+$ record and modelled concentrations in precipitation at the drilling sites agree well during the period AD 1900-1980 (Figure 5). Similarly to $NO_3^-$, there is a strong deviation after 1980 with unprecedented high $NH_4^+$ ice-core concentrations, but decreasing simulated concentrations and emission estimates in the source regions. Spain is the only country, with non-decreasing $NH_3$ emission estimates, whereas all other relevant Western European source countries show a 10-35% decrease between the 1980s and 2010s. One hypothesis to explain the discrepancy is that $NH_3$ emissions from soil amplified by increasing temperatures (Skjoth and Geels, 2013; Sutton et al., 2013) are not yet considered in the used version of the MATCH model. Another hypothesis is the nonlinearity between $NH_3$ emissions and $NH_4^+$

deposition over the Alps not fully covered by the model. The conversion of gaseous $NH_3$ to aerosol-borne $NH_4^+$ strongly depends on the presence of acidic species (i.e. $H_2SO_4$, $HNO_3$) and the pH of the aerosol. Whereas the decreasing concentrations of $SO_4^{2-}$ after the 1970s are well simulated by the MATCH (section 3.2.1.), the still high recent $NO_3^-$ concentrations observed in the ice-core composite (section 3.2.2.), which also might cause enhanced $NH_4^+$ concentrations, are not reflected by them.

Finally, also uncertain emission estimates can potentially contribute to the observed discrepancy. In any case, the still high Alpine composite $NH_4^+$ concentrations during the past decades clearly show that the introduced agricultural policies were not sufficient to create a significant downward trend in recent $NH_4^+$ levels at high altitudes in Western Europe.

In summary, the concentrations of all three major inorganic aerosol species are highly correlated between the individual sites (Figure 4). The lowest correlation coefficients are observed for $NH_4^+$ ($0.86 < r < 0.99$), due to the stronger pre-industrial

variability related to the potential contamination issue for $NH_4^+$. Highest correspondence is obtained between the different $NO_3^-$ ($0.96 < r < 1.00$) and $exSO_4^{2-}$ records ($0.92 < r < 1.00$). These results demonstrate that longer-term changes in ice-core concentration records of major inorganic aerosol species from individual sites in the European Alps provide a spatial representative signal of anthropogenic pollution trends from Western European countries.

### 3.2.2 Trends in black carbon

The CG15 rBC record and FH EC record correspond well in their general trend (Figure 2). The difference in absolute concentrations of approximately a factor of 4 was expected, since thermo-optical methods to determine EC produce consistently higher values compared to methods utilizing the light-absorbing properties of BC (Currie et al., 2002; Bond et al., 2007; Lim et al., 2014; Sigl et al., 2018). The rBC and EC concentration trend is marked by low pre-industrial levels, an increase during the second half of the 19[th] century, maximum values in the beginning of the 20[th] century and subsequent

decreasing levels. Pre-industrial rBC and EC sources are anthropogenic or natural biomass burning (Bond et al., 2007; Sigl et al., 2018). The significant increase of rBC and EC during the second half of the 19[th] century is concurrent with the rise in $exSO_4^{2-}$ concentrations and related to emissions from coal burning. The onset of substantial anthropogenic BC emissions from coal burning in Western Europe exceeding pre-industrial levels was during the 1870s, as determined with TOE of the Alpine composite (Figure 5). Coal burning emissions peaked in the first half of the 20[th] century. The still relatively high rBC and EC

concentrations after the 1950s are due to emission from burning of other fossil fuels, such as petroleum (industrial sector and traffic) and domestic emissions (Sigl et al., 2018).

Modelled atmospheric BC mixing ratios at the FH site (FLEXPART model) (Fang et al., 2019) resemble Alpine ice-core concentrations exceptionally well during the past 150 years. The Alpine composite record together with modelled mixing ratios illustrate that BC pollution in Western Europe in the beginning of the 21[th] century was again as low as at the end of the 19[th]

century demonstrating the effect of the substitution of coal with petroleum products (lower emission factors) and of the implemented air pollution control measures such as the requirement of filter systems and certified combustion efficiency and general regulations regarding firing.

Generally, a high correspondence between the individual 10-year BC records is observed (0.86 < r < 0.97) (Figure 4), illustrating that longer-term concentration variations of each single record representatively capture the Western European pollution history for BC.

### 3.2.3 Trends in trace species (Cd, Pb, F⁻)

**Cadmium**

The trends of the two Cd records from CG03 and CDD are in excellent agreement for the overlapping period AD 1890-2000 (Figure 2). Cd concentrations remain on a stable low level until the end of the 19$^{th}$ century, start to increase at the beginning of the 20$^{th}$ century and peak in the 1970s. Cd sources during the pre-industrial period are mineral dust and mining activities. Rising values from the beginning of the 20$^{th}$ century on are attributed to emissions from Zn and Cu smelters and coal burning. These extensive emissions are responsible for the exceeding of pre-industrial Alpine composite Cd levels since the 1900s (Figure 5). The major Cd source during the 20$^{th}$ century was growing emissions from Zn smelting followed by Cu smelting, whereas emissions from coal combustion became less important towards the second half of the 20$^{th}$ century (Legrand et al., 2020). Decreasing Cd concentrations after the 1970s reflect the introduction of air pollution control measures, such as the implementation of filters to clean the vent gases of Zn and Cu smelters (Legrand et al., 2020).

We find a good correspondence in the trend between the Alpine composite Cd record and estimated Cd deposition until the 1960s (Figure 5). Past Cd deposition fluxes were calculated from emissions estimates in European countries weighted by the FLEXPART emission sensitivities (combined emission scenarios (1) and (2) from (Legrand et al., 2020)). Whereas model estimates strongly decrease after the 1960s, ice-core Cd peaks in the 1970s, suggesting that the applied model emission reductions, considering the introduction of pollution control devices and other technological improvements, was too optimistic. However, the effectivity of the introduced abatement measures can clearly be detected in the ice-core records with similar Cd levels in the 1990s as in the 1920s.

**Lead**

Three Pb records from the CG (CG03, CG13, CG95) correspond exceptionally well, revealing low values until the end of the 19$^{th}$ century, with elevated levels at the beginning of the 20$^{th}$ century, strongly rising concentrations with a maximum in the 1970s, followed by a declining trend (Figure 2). This correspondence in trace element concentration records in the lower ppb range is remarkable, since the three CG cores were sampled and analysed in three different laboratories, using different methods of analyses. The CDD Pb concentrations show a similar trend to the CG cores after AD 1940. The relative elevated Pb levels in the earlier period from around AD 1890-1940 observed in the CDD core still remain unexplained. Pb emissions during the pre-industrial period were mainly from soil dust and mining activities in Europe (Schwikowski et al., 2004). During the second half of the 19$^{th}$ century and first half of the 20$^{th}$ century, emissions from nonferrous metal production, iron and steel manufacturing, and coal combustion became dominant. The TOE analysis demonstrates that already from the 1870s on the

Alpine composite Pb record exceeds the pre-industrial Pb levels, related to these emissions (Figure 5). The increasing trend in Pb concentrations after the 1940s until the 1970s resulted from the use of lead additives in gasoline, whereas the subsequent decrease after the 1970s is related to the ban of leaded gasoline (Legrand et al., 2020).

Trends of the Alpine composite Pb record and the calculated Pb deposition based on emissions estimates in European countries weighted by the FLEXPART emission sensitivities (emission scenarios (1) and (2) from (Legrand et al., 2020)) agree remarkably well (Figure 5). Ice-core records together with deposition estimates illustrate that, mainly as a result of the ban of leaded gasoline, Pb pollution in Western Europe in the beginning of the $21^{th}$ century was reduced to pre-industrial values.

**Fluoride**

The $F^-$ concentration record from FH shows low values during the $18^{th}$ and $19^{th}$ century and increasing levels from the first half of the $20^{th}$ century on. The three records from CDD, FH, and GG consistently reveal a significant rise during the 1930s-1940s peaking in the 1960s and a strong drop afterwards (Figure 2). $F^-$ sources during the pre-industrial period are mainly soil dust emissions. In the 1930s the Alpine composite $F^-$ record exceeds for the first time the variability of the pre-industrial period (Figure 5), marking the onset of substantial $F^-$ emissions from aluminium smelters in Western Europe. Thus, from the 1930s to the 1980s the main contributor to anthropogenic $F^-$ changes at CDD, FH, and GG was HF emissions from the aluminium industry in France, Germany, and Switzerland, with the main smelters located in the Auvergne-Rhône-Alpes and in the Swiss Rhone valley (Eichler et al., 2000a; Preunkert et al., 2001b). The emission of huge amounts of HF from this source caused severe ecological damage during the 1960s and 1970s (Eichler et al., 2000a). The strong drop of $F^-$ concentrations after the 1960s was the effect of installing waste-air purification system in aluminium smelters, such as cap systems equipped with $Al_2O_3$ filter to chemisorb HF (Preunkert et al., 2001b). During the $20^{th}$ century, coal burning contributed to anthropogenic $F^-$ changes, with a share of less than 30% (Preunkert et al., 2001b).

The Alpine composite $F^-$ record and HF emission estimates from the aluminium industry of France and Switzerland during the period AD 1920-1980 show comparable trends (Figure 5). Only in the 1990s the Alpine $F^-$ level is elevated compared to the emission estimates, pointing to an increasing share of other anthropogenic sources like cement and phosphate industrial processes (Preunkert et al., 2001b). However, the efficacy of the introduced abatement measures in Western Europe mainly for the aluminium industry, is well reflected in the ice-core records, revealing $F^-$ levels in the 1990s similar to the previously lower levels in the 1950s.

Similar to the findings for the major inorganic aerosol species and BC, also 10-year concentration records of all three trace species are highly correlated between the individual sites (Figure 4). Lower correlation coefficients are in part observed for Pb ($0.84 < r < 0.98$), related to the relative elevated CDD Pb levels in the period AD 1890-1940, whereas highest correspondence is obtained between the individual Cd and $F^-$ records ($0.91 < r < 0.99$). We conclude that also for trace elements, a single Alpine ice core can be a representative archive of Western European air pollution trends.

## 4 Summary and Conclusion

Here we present the first study considering longer-term records (AD 1750-2015) of seven anthropogenic pollutants from four different high-Alpine ice-core sites in the European Alps (CDD, CG, FH, GG). The systematic comparison between ice-core records of BC, Cd, $F^-$, $NH_4^+$, $NO_3^-$, Pb, and $exSO_4^{2-}$ allowed addressing the question of how representative one ice core is with respect to the reconstruction of atmospheric composition in the source region.

High-Alpine ice-core sites have the advantage that they are located in less than 100 km distance, with altitude differences of < 500 m. Furthermore, the locations are either not affected by melting processes (CDD, CG) or melt-disturbed parts were removed before analyses (FH, GG). Emission sensitivities produced with the atmospheric aerosol transport model FLEXPART give evidence that the considered ice-core sites receive air masses from the same source region of pollutants, i.e. from Western European countries surrounding the Alps (Switzerland, France, Italy, Germany, Austria, Spain, and Slovenia).

In general, pollution concentration levels are similar at the three sites CG, FH, and GG. However, CDD concentrations of all studied species are on average about a factor of 1.4 (1.1-1.7) lower, most probably caused by a higher share of winter precipitation characterised by low concentrations. Thus, absolute pollution concentrations at the Alpine sites can vary depending on the prevailing seasonal distribution of accumulated precipitation, suggesting that they do not provide a spatial consistent signal.

In contrast, longer-term concentration variations of all investigated air pollutants feature a uniform timing in species-dependent anthropogenic impact at the four sites. Our results demonstrate that all ice-core records from the different sites in the European Alps provide a representative signal of anthropogenic pollution changes in Western European countries. Thus, in our study we show for the first time, how consistent this pollution history is recorded in different Alpine ice cores.

We were able to refine the history for different compounds by pinning down their onset of pollution, maxima, and recent changes. Based on the time of emergence analyses on the obtained Alpine ice-core composite records the onset of significant air pollution exceeding pre-industrial levels (AD 1750-1850) in Western Europe was pinpointed to the 1870s and 1880s for BC, $exSO_4^{2-}$, Pb, and $NH_4^+$, mainly caused by emissions from coal combustion and agriculture, respectively. Cd, $F^-$, and $NO_3^-$ concentrations in the atmosphere predominantly from emissions of zinc and aluminium smelters and traffic, respectively, started surpassing pre-industrial values later; in the first half of the 20$^{th}$ century for Cd and $F^-$, and during the 1960s for $NO_3^-$.

Consistent pollutant-dependent maxima and recent changes at the four sites reveal a divers efficiency of implemented emission abatement measures. BC concentrations have a maximum in the first half of the 20$^{th}$ century, whereas those of Cd, $exSO_4^{2-}$, $F^-$, and Pb peak during the 1970s. Concentrations of all five pollutants significantly decrease afterwards, related to the desulphurisation of coal, the use of low-sulphur fuels, the introduction of filters and scrubbers in power plants and metal smelters, and the ban of leaded gasoline for improving the air quality in Western Europe. In contrast, the ice-core concentrations of the nitrogen species $NO_3^-$ and $NH_4^+$ in the beginning of the 21$^{th}$ century are unprecedented in the context of the past 250 years, indicating that the introduced measures, such as denoxification of industrial and vehicle emissions or $NH_3$

reductions in fertilizer industry and livestock breeding were still not sufficient to have a major effect on reducing recent $NO_3^-$ and $NH_4^+$ levels at high altitudes, respectively.

Ice-core records of four out of seven investigated pollutants (BC, $F^-$, Pb, $exSO_4^{2-}$) are in good correspondence with model trends of either their emission history, atmospheric concentrations, or deposition data. This is not the case for the recent, still high concentrations of $NH_4^+$ and $NO_3^-$ and the Cd concentration maximum in the 1970s. Hypotheses for the disagreement are potential uncertainties of the emission estimates, or model misrepresentation of partitioning processes between the gas and particle phase (e.g. for $NH_4^+$ and $NO_3^-$) and of chemical reactions controlling the atmospheric lifetime. The fact that only four

of seven pollutant records do agree with available model estimates, illustrates the necessity to include such ice core-based reconstructions for model constrain.

**Acknowledgements**

Tinja Olenius (Swedish Meteorological and Hydrological Institute) is acknowledged for performing the MATCH-BIODIV simulations in the research project BioDiv-Support. Camilla Andersson and Tinja Olenius were funded through the 2017-2018

Belmont Forum and BiodivERsA joint call for research proposals, under the BiodivScen ERA-Net COFUND programme, and with the funding organisations AKA (Academy of Finland contract no 326328), ANR (ANR-18-EBI4-0007), BMBF (KFZ: 01LC1810A), FORMAS (contract no:s 2018-02434, 2018-02436, 2018-02437, 2018-02438) and MICINN (through APCIN: PCI2018-093149). We thank Jacobo Gabrielli for providing the Cd data from Colle Gnifetti. Data from CG03B+CG15 ice core are available at PANGAEA, doi.org/10.1594/PANGAEA.894787

**Data availability**

The data presented in this work are archived at the NOAA (National Oceanic and Atmospheric Administration) data center for paleoclimate (ice core sites): http://www.ncdc.noaa.gov/paleo/study/xxxxx.

**Author contribution**

AE, ML, and MS conceived the study and wrote the paper. AE, TMJ, and SP contributed to data analyses and interpretation.

CA and ME performed the MATCH calculations. AP and SE performed the FLEXPART modelling. All authors contributed to manuscript preparation.

**Competing interests**

No competing interests are present.

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

**Table 1:** Details of the study sites from which the ice cores used in this study are recovered.

| Study site | Country | Year of drilling | Abbreviation | Coordinates | Altitude m asl | Accumulation m w.eq./year | Reference |
|---|---|---|---|---|---|---|---|
| Colle Gnifetti | CH | 1982 update 1995 | CG95 | 45°53'33''N; 07°51'52''E | 4450 | 0.33 | Schwikowski et al. (1999b, 2004) |
| | | 1982 update 1993 | CG93 | 45°55'43''N; 07°52'30.4''E | 4470 | 0.22 | Wagenbach et al. (2012) |
| | | 2003 | CG03 | 45°55'50''N; 07°52'33''E | 4455 | 0.46 | Jenk et al. (2009); Sigl et al. (2018) |
| | | update 2015 | CG15 | | | | |
| | | 2013 | CG13 | 45°55'44.2''N; 7°52'34.6''E | 4450 | 0.22 | Hoffmann et al. (2018) |
| Grenz-gletscher | CH | 1994 | GG | 45°55'28''N; 07°52'3''E | 4200 | 2.7 | Eichler et al. (2000) |
| Fiescher-horn | CH | 2002 | FH | 46°33'3''N; 08°04'0.4''E | 3900 | 1.4 | Jenk et al. (2006) |
| Col du Dôme | F | 1994 | CDD | 45°50'28''N; 06°50'52''E | 4250 | 2.4 | Preunkert et al. (2000) |
| | | 2004 | | | | | Legrand et al. (2013) |
| | | 2012 | | | | | Legrand et al. (2018) |


**Table 2:** Origin of the datasets for the different ice cores from CDD, CG, FH, and GG.

| Species | CG | GG | FH | CDD |
|---|---|---|---|---|
| $NH_4^+$, $NO_3^-$, $SO_4^{2-}$ | CG03A+CG15 (Sigl et al., 2018), CG93 (Wagenbach et al., 2012) | (Eichler et al., 2000b; Eichler et al., 2004) | Jenk (2006) | (Preunkert et al., 2001a; Preunkert et al., 2003; Fagerli et al., 2007) |
| EC, rBC | rBC CG03B+CG15 (Sigl et al., 2018) | | EC (Sigl et al., 2018) | |
| F- | | Eichler et al. (2000a) | Jenk (2006) | Preunkert et al. (2001b) |
| Pb | CG03 (Gabrieli and Barbante, 2014) CG13 (More et al., 2017) CG95 (Schwikowski et al., 2004) | | | Legrand et al. (2020) |
| Cd | CG03 (Gabrieli, 2008) | | | Legrand et al. (2020) |


**Table 3:** ExSO$_4^{2-}$ concentrations for different 10-year periods derived from ice cores and the MATCH model (MATCH-ECLAIRE and MATCH-BIODIV datasets). Unit is ppb (corresponding to µg/l).

| exSO$_4^{2-}$ (ppb) | Ice core 1970-1979 | MATCH-ECLAIRE 1970-1979 | Ice core 1990-1999 | MATCH-ECLAIRE 1990-1999 | MATCH-BIODIV 1990-1999 |
|---|---|---|---|---|---|
| CG | 827 | 7258 | 524 | 4113 | 696 |
| GG | 876 | 7258 | | 4113 | 696 |
| FH | 743 | 5189 | | 2889 | 696 |
| CDD | 495 | 2310 | 336 | 1291 | 672 |


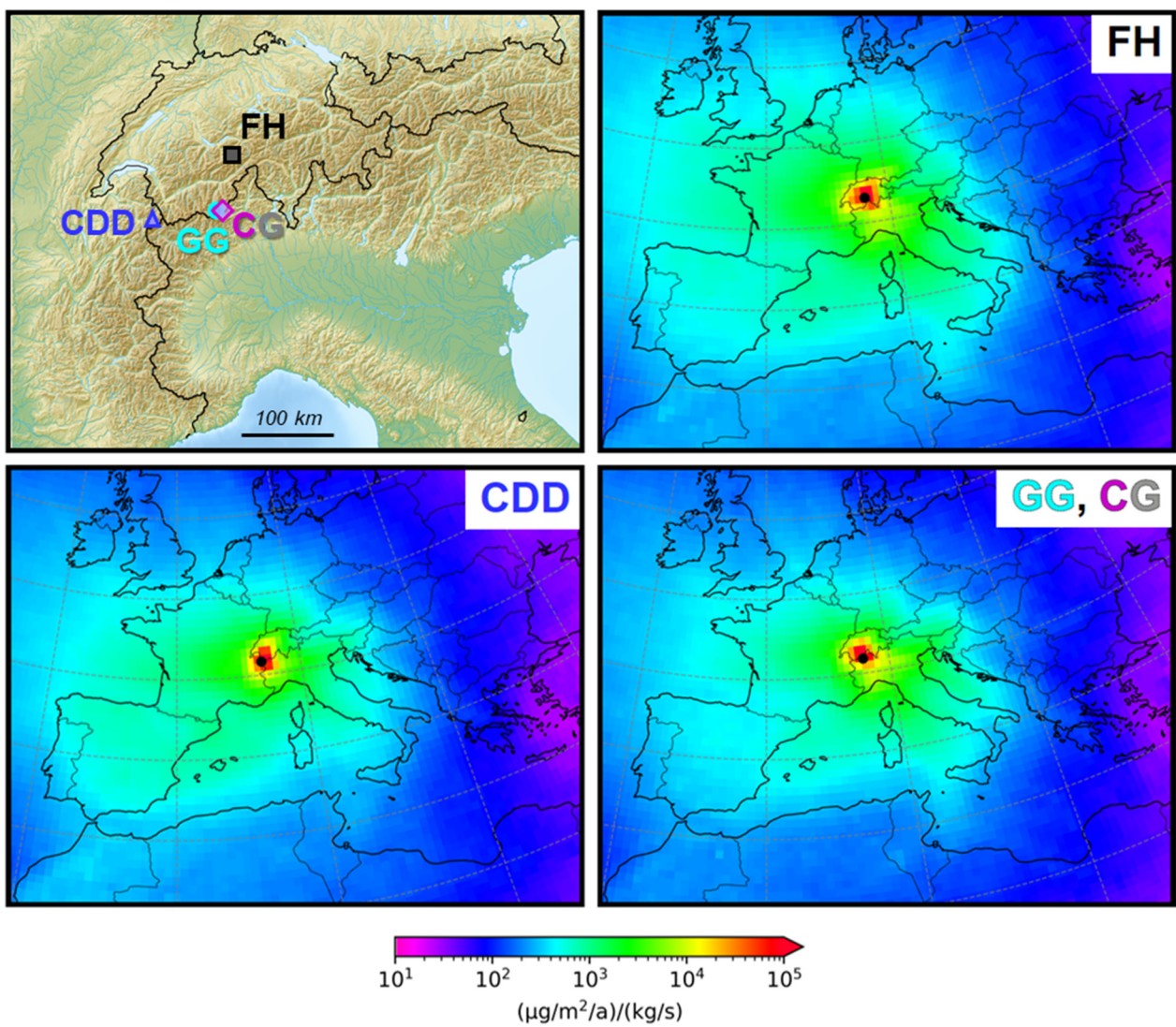

**Figure 1:** Topographic map of the European Alps (Credit Lencer/Pechristener, Alps location map borders, CC BY-SA 3.0, http://creativecommons.org/licenses/by-sa/3.0 or GFDL, http://www.gnu.org/copyleft/fdl.html, via Wikimedia Commons (2010), Date of access: 23/06/2022) together with the four ice-core sites (upper left panel). Average $SO_4^{2-}$ emission sensitivities (in µg m$^{-2}$ a$^{-1}$/kg s$^{-1}$) at the FH, CDD and GG/CG site (black dots) based on FLEXPART model simulations (0.5x0.5°) for the period AD 2000-2009 (upper right and lower panels).

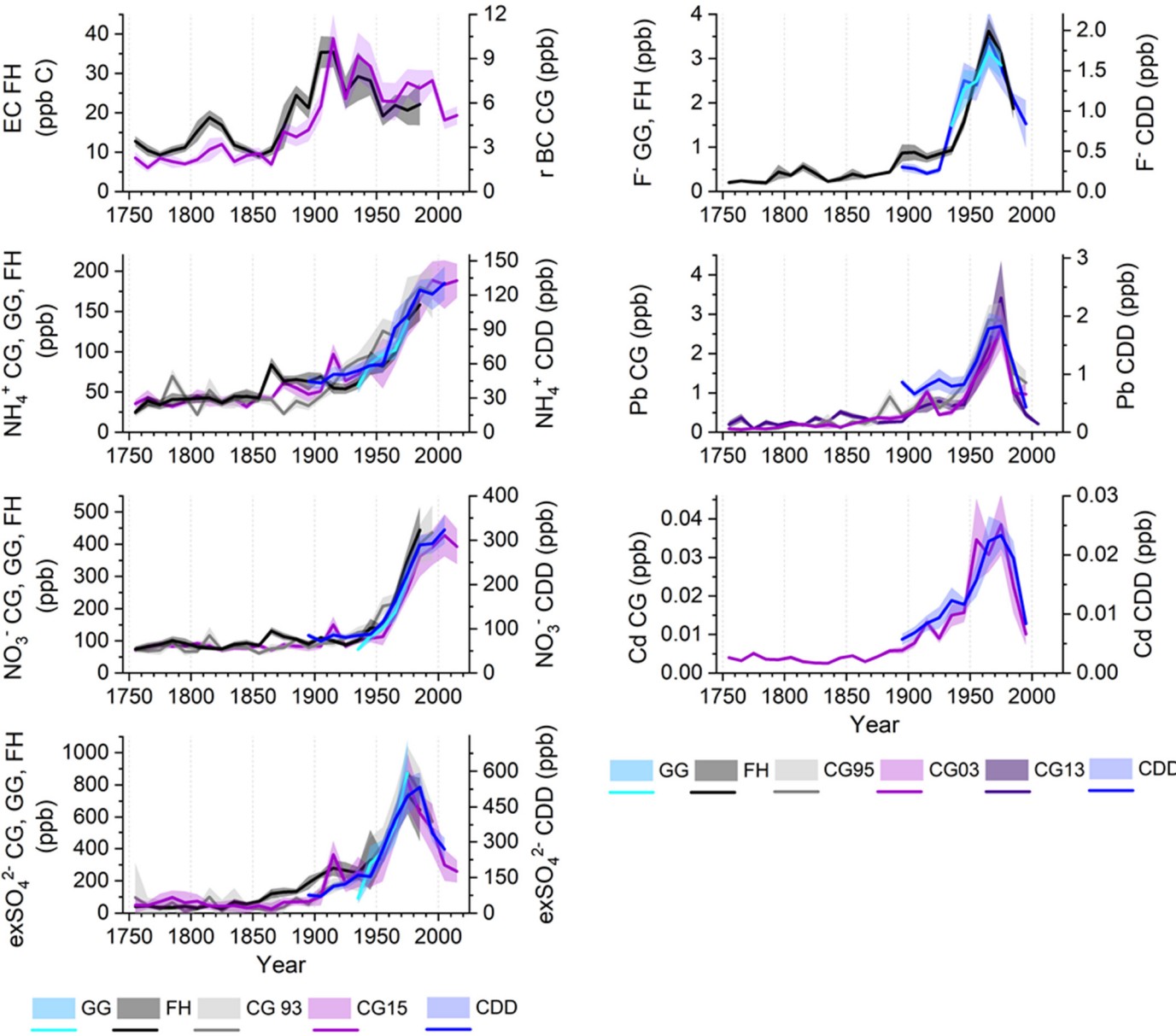

**Figure 2:** Compilation of decadal ice-core concentration records (mean ±1 standard error) from the four ice-core sites (CDD, CG, FH, GG) for BC (EC and rBC), major inorganic aerosol species $NH_4^+$, $NO_3^-$, $exSO_4^{2-}$, and trace species $F^-$, Pb, Cd covering the time period 1750-2015.

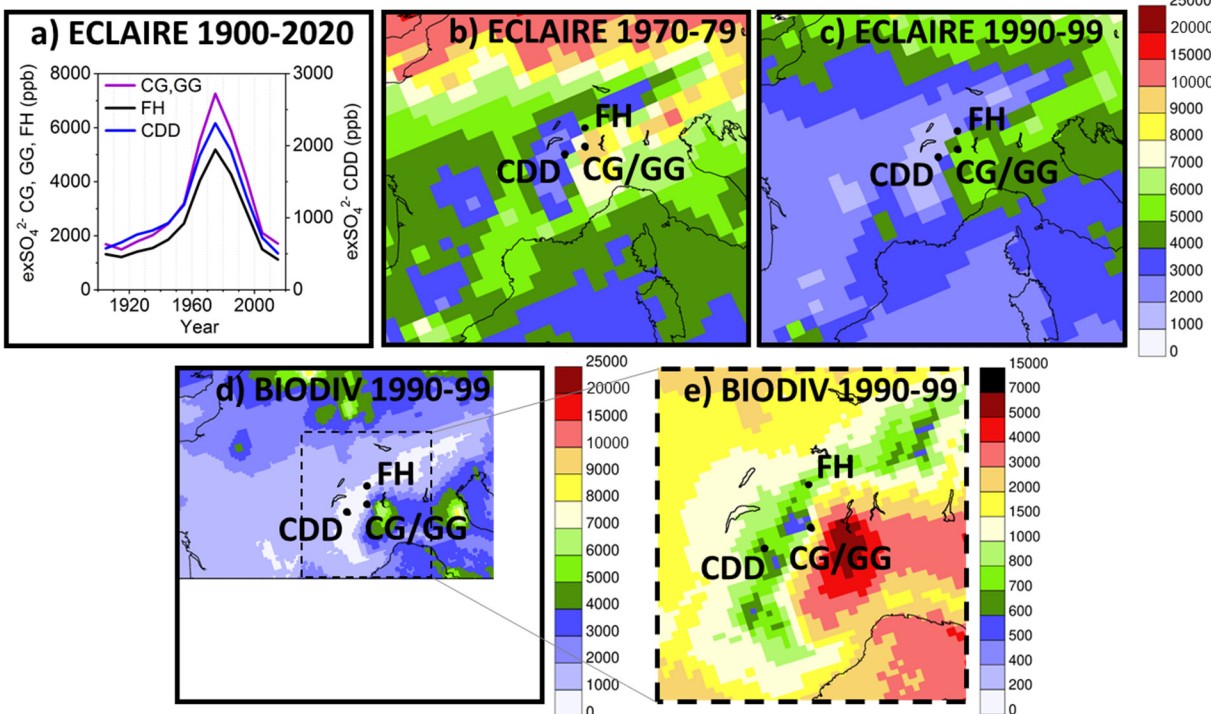


**Figure 3:** a) Modelled exSO$_4^{2-}$ concentration records at the four ice-core sites CDD, CG/GG and FH using the MATCH model (MATCH-ECLAIRE data set, period AD 1900-2020) and b-e) comparison of spatial exSO$_4^{2-}$ concentrations in precipitation at the surface (ppb) obtained by the MATCH model (b) MATCH-ECLAIRE data set AD 1970-1979, c) MATCH-ECLAIRE data set 1990-1999, d),e) MATCH-BIODIV data set 1990-1999). Note that color codes and covered area are identical for panels b)-d), whereas panel e) represents a magnified
inset of panel d) with a different color code to illustrate variations in the high-Alpine regions.

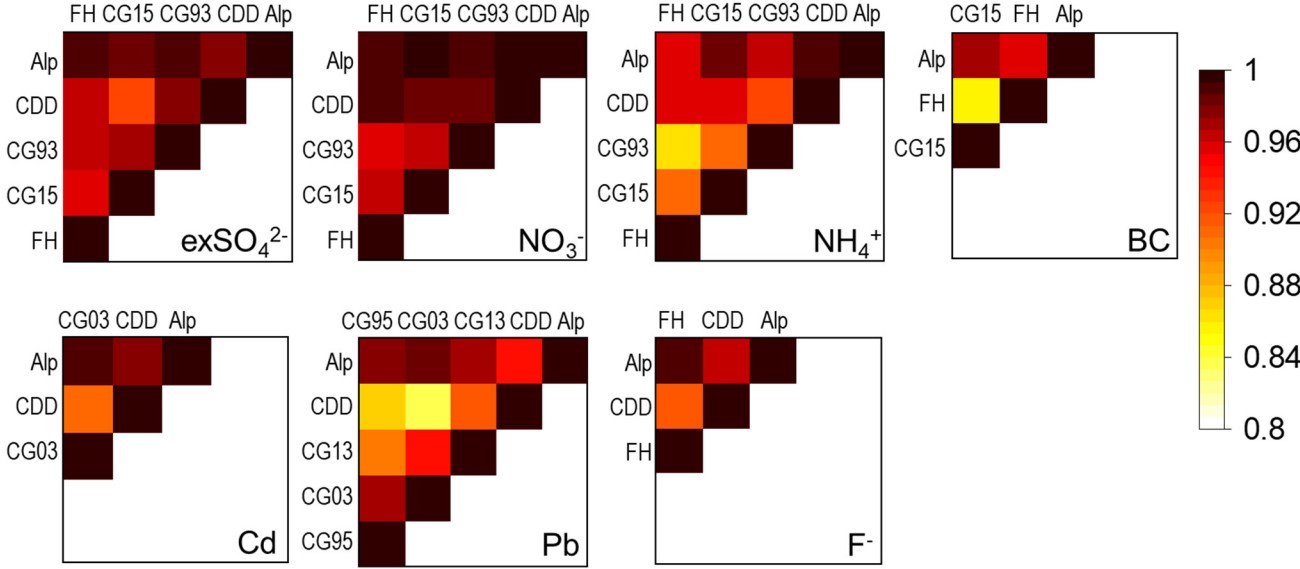

**Figure 4:** Correlation matrices showing the correlation coefficients r between 10-year concentration records from different ice cores and the Alpine composite (Alp) for individual pollutants exSO$_4^{2-}$, NO$_3^-$, NH$_4^+$, BC, Cd, Pb, and F$^-$.


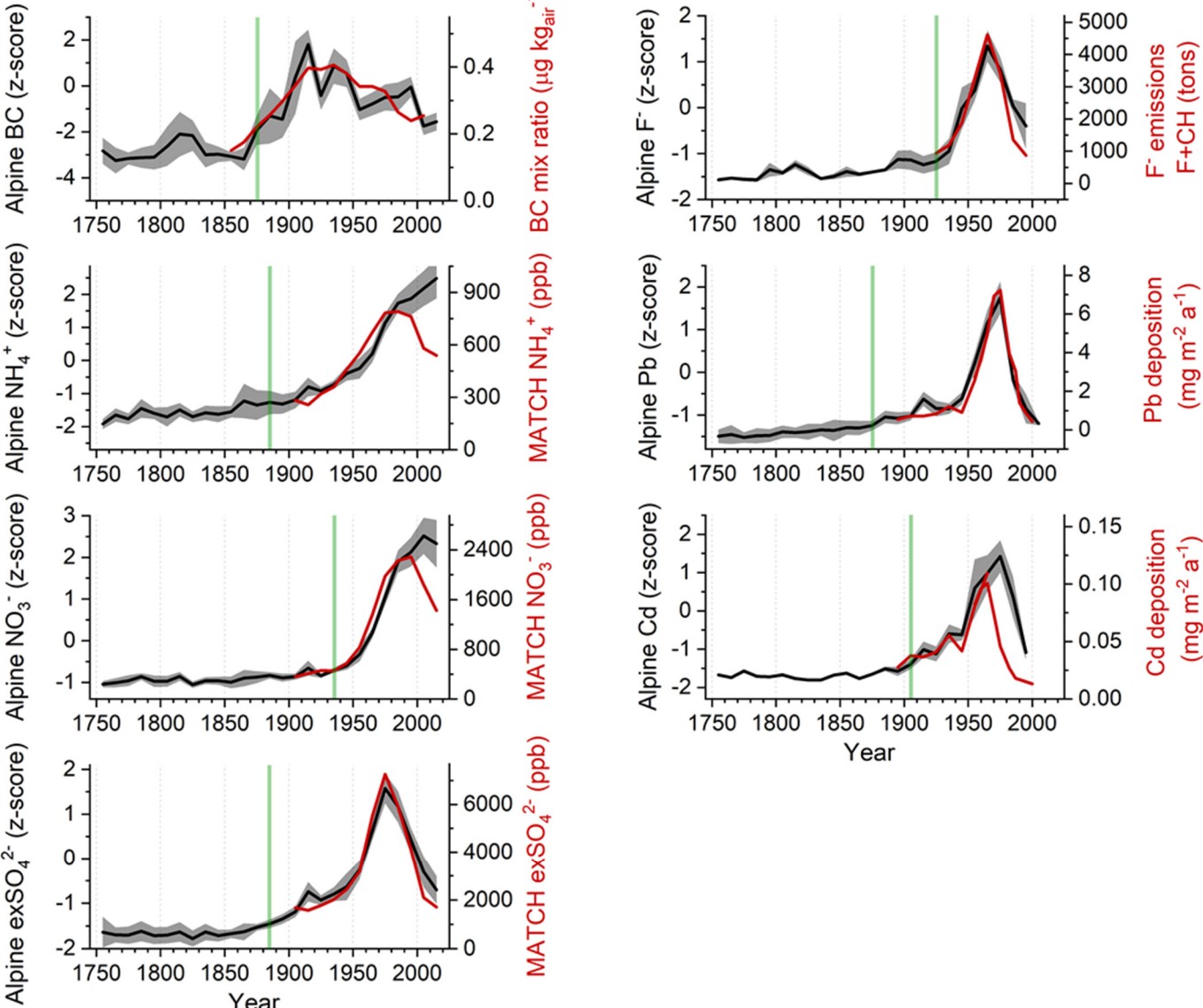

**Figure 5:** Compilation of decadal Alpine ice-core composite z-scores (mean ±1 standard error) based on longer-term concentration records of BC, major inorganic aerosol species $NH_4^+$, $NO_3^-$, $exSO_4^{2-}$, and trace species $F^-$, Pb, Cd from CDD, CG, and FH for the time period AD 1750-2015 (black/gray curves). The TOE (time of emergence, green line) is additionally shown together with modelled concentrations of $NH_4^+$, $NO_3^-$, $exSO_4^{2-}$(MATCH-ECLAIRE data set) (Engardt et al., 2017), estimated BC air mixing ratios (Fang et al., 2019), $F^-$ emission estimates of France and Switzerland (Preunkert et al., 2001b), and Pb and Cd deposition estimates (Legrand et al., 2020) (red lines).