# Peer review of "Consistent histories of anthropogenic Western European air pollution preserved in different Alpine ice cores"

_The Cryosphere, 2022_

## Author Comment (AC1)

This study combines and discusses air pollution records reconstructed from four different ice-core sites in the European Alps. My comment suggests some technical clarifications concerning the ice-core records from Colle Gnifetti which I invite the authors to address in the revision.

Thanks Michael for the suggested technical clarifications. We will add them to the manuscript accordingly.

L 142: Please add as citation for the dating Sigl et al., (2018) providing necessery details about the individual ice cores (Table 1) and the final dating (Supplementary) used to derive the NO3, NH4, rBC and SO4 records used for this study together with access to the underlying high-resolution datasets (https://doi.org/10.1594/PANGAEA.894787).

The reference Sigl et al. (2018) is already included in Tables 1 and 2, but we will also add it to line 142.

L 172: Are these new IC analyses for the CG15 ice core extending back to 1750; or are these the upper 12m of CG15 merged with the records from the two CG03 ice cores (CG03A and CG03B)? Please clarify; see Table 1 in Sigl et al., (2018) for details. If these are new analyses, how was the dating of CG15 done, and how do the records compare relative to the CG03 records during the period of overlap? If the latter, which records did you use for your analyses? CG03A, CG03B or a stacked record of both?

These are the records from the upper 12m of CG15 merged with those from the CG03A core. Details will be added to Tables 1 and 2.

L 187: Same question here: Are these new rBC analyses from the CG15 ice core? The reference to Sigl et al., (2018) suggests these are from CG03B.

These are the records from the upper 12m of CG15 merged with those from the CG03B core. Details will be added to Tables 1 and 2.

L 196: This now answers the two questions noted above. To avoid misunderstanding, I suggest to correct or clarify the meaning of the core labels in the sections above and in Table 1 and 2, since the CG15 ice core is actually encompassing the time period 1750-2015 overlapping with CG03 ice cores.

Details will be added to Tables 1 and 2.

L 487: Add "Data from CG ice cores are available at PANGAEA, doi.org/10.1594/PANGAEA.894787" to acknowledge the work generating these measurements.

We will add this sentence to the final version.

Table 2: Suggest to replace "CG15" with "CC03_CG15" or similar, to show this is the CG03 ice core updated by the CG15 ice core.

For the clarity of the figures, we prefer not to have overlong ice core denotations. Therefore we keep the "CG15", but clarify details in Tables 1 and 2.

---

## Author Comment (AC2)

We would like to thank referee#1 for the willingness to review our manuscript. We really appreciate the detailed comments and suggestions of the referee. They helped to improve the clarity of the manuscript. Please find below our responses to the comments (in blue) and our changes to a potential revised manuscript (in blue, but italic)

General comments

In this study the authors present a synthesis of air pollution data from four different High Alpine European ice cores for the last ~250 years. They analyse how representative the single core data are for the common pollution history of the area. They create composite records of 10-year averages for the respective pollution species and calculate "times of emergence" (TOE) when the actual anthropogenic pollution becomes significant. This is an interesting topic and especially the investigation of the representativity of one ice core record for the atmospheric signal of an entire region is very important and was not sufficiently discussed so far. Also the findings of the different times of emergence of the pollution signals in Europe and the present day conditions, including partly failed abatement measurements, are highly relevant and should be published.

The paper is overall well written. The paragraphs follow a clear structure and are easy to follow. However, the general focus of the study remains a little unclear to me. The title and the statements at the end of the discussion (section 3) paragraphs rather suggest that the investigations of the TOE and the evaluation of the present day situation were the main objectives. On the other hand, the authors also dedicate a good deal of the study to the evaluation of the representativity of a single core record for the whole region. In my opinion, both discussions lack depth and critical evaluation. The paper would definitely benefit from a clarification of the main objective and a more focused approach (methodological or interpretation of pollution history). Especially the discussion section (3.2-3.4) is sometimes rather vague and needs some revision. I think the discrepancies between the single core records (Fig. 2) and the resulting consequences for the composite record (and the following interpretation) are not discussed and assessed sufficiently (see specific comments).

The main object of the study is to investigate, how representative one Alpine ice core is to document the pollution history of Western Europe. As suggested by the referee we will clarify this throughout the manuscript, where not done already (end of introduction, first chapter of section 3., conclusion). We think that the chosen title does not contradict the main object of the study and prefer to keep it. In chapter 3 we will clarify that for investigating the representativeness a) absolute concentration levels and b) long-term concentration trends of BC, Cd, F$^-$, NH$_4^+$, NO$_3^-$, Pb, and exSO$_4^{2-}$ from the four sites (CDD, CG, FH, and GG) were compared. For a more quantitative and less "vague" comparison of the long-term trends we added a correlation analyses (Fig. R1).
According to the comments of the referee we will rearrange the structure of the Results and Discussion part:
- 3. Results and Discussion
general paragraph:
*"Average emission sensitivities during the period 2000-2009 derived with FLEXPART, indicate that in general the four ice-core sites CDD, CG, FH, and GG are most sensitive to emissions from the countries surrounding the Alps, i.e. Switzerland, France, Italy, Germany, Austria, Slovenia, and Spain, without substantial discrepancies between the four sites (Figure 1). A noticeable extension of the pollution source area into Spain is recognized for the westernmost CDD, and into larger parts of*

*Germany for the northernmost FH site.*

*To investigate, how representative ice-core records from the four sites are to document the pollution history in this deduced source area we systematically compared a) absolute concentration levels (section 3.1.) and b) long-term concentration trends of BC, Cd, F⁻, NH₄⁺, NO₃⁻, Pb, and exSO₄²⁻ (section 3.2.).*"

- 3.1. Absolute concentrations
  (former chapter 3.1.2)
- 3.2. Trends
general paragraph:
*"Generally, an excellent agreement in the ice-core long-term concentration trends of all investigated pollutants is observed for CDD, CG, FH, and GG (Figure 2). This is confirmed by correlation analyses, revealing high correlation coefficients ($0.86 < r < 0.99$) between different records of the same compound (new figure R1). All correlations are significant at the 0.001 level.*
*The observed, characteristic differences between the concentration trends of the individual seven pollutants reflect dissimilarities in respective emission sources, emission trends, atmospheric chemistry of precursor species, and compound specific implementation and efficiency of emission abatement measures, which is discussed in detail in the sections 3.2.1-3.2.3. For individual pollutants this additionally includes discussion of common features and differences in the 10-year concentration records of the four sites and the consequences regarding the formation of Alpine composites and representativeness to capture the Western European pollution history. The latter is evaluated by TOE and comparison with available model estimates to precisely time the onset of increased pollution levels and model validation of air pollution trends, respectively."*
- chapters 3.2.1 – 3.2.3.
former chapters 3.2.-3.4., +discussion about consequences of common/different features for the formation of Alpine composites and representativeness will be added also considering the results of the correlation analyses

[Figure]

Fig. R1: Correlation matrices showing the correlation coefficients r between 10-year concentration records from different ice cores and the Alpine composite (Alp) for the individual pollutants exSO₄²⁻, NO₃⁻, NH₄⁺, BC, Cd, Pb, and F⁻

My other major point of criticism is the methodological approach for averaging in section 2.3. It is not justified why the authors calculate 10-year averages for the single ice core records. Is that indeed necessary to "average out inter-annual-fluctuations"? (L203). If only looking at a 250 year time range this seems to smooth out a lot of variation. Could another averaging window also be sufficient and maybe provide more detailed insights? This needs some statistical evaluation. The reasoning behind the 10-year averaging seems random, remains unclear so far and will have consequences for the following discussion of TOE and pollution records. This section needs to be extended, clarified and methodologically justified.

We agree with the referee that this point of the chosen averaging window was not explained properly and we will extend the discussion in section 2.3 and add a new supplementary figure R2. There are two main reasons for the chosen 10-year window: (1) averaging out inter-annual fluctuations mainly related to variations in pollutants vertical transport to the high altitudes controlled by atmospheric stability and (2) accounting for the low data resolution and high dating uncertainty in the lower ice core parts (e.g. 10-25 years for the FH in the period 1750-1800).

[Figure]

Fig. R2: Comparison of CG15 exSO$_4^{2-}$ concentration records with CG15 $\delta^{18}$O records (panels a-c) and with modelled exSO$_4^{2-}$ concentrations (MATCH-ECLAIRE) (panels d-f) in the period 1920-2015 using different averaging windows (a,d – 1-year, b,e – 5 year, c,f – 10 year).

To select an adequate averaging period, we investigated the influence of the different averaging windows on the CG15 exSO$_4^{2-}$ record in the period 1920-2015 (Fig. R2). Figure R2a illustrates that

annual exSO$_4^{2-}$ maxima/minima are in part synchronous with $\delta^{18}$O maxima/minima. Since $\delta^{18}$O is a temperature proxy (Bohleber et al., 2013), this reflects a stronger/weaker vertical transport of pollutants from their source regions at lower elevations to the high-altitudes sites during warmer/colder summers. This is still partially visible in the 5-year means (Fig. R2b), e.g. the temperature ($\delta^{18}$O) maximum in the period 1970-75 lead to an amplification of the exSO$_4^{2-}$ concentration at CG compared to what is expected based on the SO$_2$ emission history (Fig. R2e). This modulation of the ice-core record by year-to-year variability of vertical transport/atmospheric stability also produces a larger variability compared to the modelled concentrations for the annual and 5-year means (Fig. R2 d,e), but not for the 10-year averages. The MATCH-ECLAIRE model does not have sufficient resolution to resolve vertical transport.

Thus, we argue that the ice-core 10-year averages (1) are representative for changing pollutant emissions, averaging out the inter-annual (short-term) fluctuations related to temperature dependent pollutant vertical transport to the high-alpine sites and (2) take into consideration the low data resolution and higher dating uncertainty in the pre-industrial period.

I have some specific comments, especially about some figures and the respective discussions.

Specific comments

L21: please put "long term" in perspective. For most ice core studies, 250 years would not be considered long term

We put the wording "long-term" already in perspective by adding the time period (AD 1750-2015) and think this is sufficient information for the abstract of the paper.

L21: please specify "several" ice cores

Will be specified: *"Two to five ice cores from four high-Alpine sites located in the European Alps analysed by different laboratories were used, depending on the data availability of the different air pollutants."*

L37f: This statement lacks evaluation. What are the consequences from only four core records corresponding well with the modelled trends? Please elaborate

We will add potential reasons: "*Only four ice-core composite records (BC, F$^-$, Pb, exSO$_4^{2-}$) of the seven investigated pollutants correspond well with modelled trends, suggesting uncertainties of the emission estimates or in the representation of chemical reactions in the model for the other pollutants.*"

L51: change "ice cores" to "ice core records"

We will change this accordingly.

L66ff: Maybe a problem of wording: "… indicates common source regions for the different Arctic sites…" I think this is a rather generalized and bold statement, especially because in the next sentence you state that this is not true for e.g. Svalbard. Please clarify.

Will be changed to *"This finding indicates common source regions for the different Greenland sites in North America and Europe…"*

L79f: Reference for huge Himalayan dust deposition?

We will add a reference (Sierra Hernandez et al., 2018) accordingly.

L87: you could add: Preunkert, S., McConnell, J. R., Hoffmann, H., Legrand, M., Wilson, A. I., Eckhardt, S., et al. (2019). Lead and antimony in basal ice from Col du Dome (French Alps) dated with radiocarbon: A record of pollution during antiquity. Geophysical Research Letters, 46, 4953– 4961. https://doi.org/10.1029/2019GL082641

We will add this reference.

L90f: The sentence about the downward trend is somewhat repetitive throughout the manuscript. Please shorten.

Will be shortened and repetitions avoided.

L93: Please specify that "annual" concentrations at Colle Gnifetti are also heavily biased towards the summer season because of major wind erosion of winter snow.

This point is already detailed in the chapter 2.1.1.

L102: which were the "different laboratories"?

This is different for specific pollutants and is already mentioned in Chapter 2.2.

L114: Can you add a reference for the atmospheric lifetime of the species? Are they the same for all different pollutants?

Atmospheric lifetimes vary depending on the conditions (occurrence of precipitation). We think for the purpose of this study, an average lifetime in the order of a few days for all the investigated species is a reasonable statement. We will add a reference accordingly (Seinfeld and Pandis, 2016).

L137: please add: "… parts or all of the winter snow cover…"

Will be added.

L138: add per year after w.eq.

Will be added.

L139: Please weaken this statement to "… ice at bedrock can be more than…" and cite additionally: Hoffmann, H., Preunkert, S., Legrand, M., Leinfelder, D., Bohleber, P., Friedrich, R., & Wagenbach, D. (2018). A New Sample Preparation System for Micro-14C Dating of Glacier Ice with a First Application to a High Alpine Ice Core from Colle Gnifetti (Switzerland). Radiocarbon, 60(2), 517-533. doi:10.1017/RDC.2017.99

Statement will be weakened (*"and the observation that up to 15000 years old ice was found at the bedrock."*) and the suggested reference added.

L173: were the IC measurements discrete or quasi-continuous? What was the spatial resolution? Please specify.

*"on discrete samples"* will be added. The spatial resolution was different from core to core and is detailed in the different references given in table 1 and 2.

L174: Is there a reference for the IUP measurements?

The given reference (Preunkert et al., 2001a) does also include the IUP measurements.

Section 2.2.3: Are there no measurements available for the other cores? Please explain.

For this study we used records that were published until beginning of 2022. There are indeed no other published continuous measurements.

L197f: Is this sea salt and mineral dust correction valid for such rather continental sites?

Yes it is based on the evidence in the cited paper (Schwikowski et al., 1999b).

L201: It is unclear what happened after the separate calculation of summer and winter mean concentrations. Were they also evaluated separately? Please extend and clarify.

Based on the evidence given in the previous CDD papers, winter layers at the CDD site were found to generally thin with depth relative to summer layers due to changes in depositional processes upstream of the drilling site (Preunkert et al., 2000). Therefore annual values were defined as arithmetic mean of winter and summer concentrations. The separate evaluation of the summer and winter concentration records is not subject of the paper, since such a separation of summer/winter layers could not be done on other cores due to the low resolution in the pre-industrial parts. For the CDD it has been discussed in many previous studies (e.g. Preunkert et al., 2003).

L203: See general comments. How were the 10-year averages calculated? Just arithmetic mean? Weighted? Please extend and explain.

Arithmetic means were calculated, will be added.

L205: see general comments "should better reflect" is no sufficient justification for the 10-year averaging. Please assess better and explain reasoning.

See detailed answer to this general comment above.

L210: The reason for not including the GG records does not convince me. 50 years out of 250 is already 20%. If this is considered too short, the GG core should have been left out completely.

For calculating z-scores, averages and stdev of a common overlapping period have to be considered. We consider the 50 years period covered by the GG core too short (just five data points) to obtain representative values of these two measures. Following the referees comment for L282, we prefer to leave the GG records in Fig. 2 and related discussion in the manuscript to compare concentration data from this high-accumulation site (~2.7 m weq/yr) with the ones from the close-by low-accumulation CG site (~0.3 m weq/yr).

L211: Again, are there no BC /EC records available for the other cores? Clarify.

There are no other published continuous records.

L219 and Fig. 1: It is really hard to see significant differences between the FLEXPART results for the different locations. Maybe this figure could benefit from a larger zoom on the locations and / or a different colour scheme if the goal is to show the very small large-scale variations?

There are no substantial differences in the pollutant source areas for the different sites (line 246). Since the goal here is to show common general source areas in Western Europe, we prefer to leave Fig 1 as it is.

L221: "Most species" – which ones are not present as sub-micrometer aerosols? Specify.

We will change "*most species*" to "*all species are predominantly present…*"

L229: how exactly was this calculated? Please elaborate.

A detailed description of the calculation of the source-receptor relationship for deposited mass in FLEXPART is provided by Eckhardt et al. (2017). We will add a corresponding sentence at the end of the paragraph.

L234: spatial distribution referring to what region? Please specify

The MATCH model was applied to a European domain for both datasets (MATCH-BIODIV and MATCH-ECLAIRE), as indicated in the opening sentence of section 2.5. We used these simulations but focusing on the spatial variation in the Alps in this study (e.g. Fig 3). We will clarify that the modelled spatial distribution of exSO$_4^{2-}$ concentration in the Alps, used in the present study, was extracted from these simulations.

L238: which climate simulations? The whole paragraph 2.5 is rather vague. Please specify.

The climate simulations (and emission data) used for the MATCH-ECLAIRE simulations are described in the Engardt et al. (2017) paper, while the MATCH-BIODIV will be described in a forthcoming paper and in other present work. It is outside the scope of the present paper to explain all details of the atmospheric chemistry simulations here, but we will update paragraph 2.5 with more details.

L243: how do you define long-term-trend? What would happen for a smaller / larger averaging window?

We will change this term to "long-term variations" and specify this in the revised version. The justification for the choice of the 10-year averages is given above.

L254f: Please reformulate this sentence. It seems odd if you write "could be explained" when you dismiss this possibility later in the text. Also the west-east gradient seems confusing in this context, when in Fig. 1 you mention that for CDD the source region is extended to the west.

We will reformulate the sentence: "*Potential reasons for the lower CDD concentrations are…*"

L255: Do you have estimates for the ratio of the summer / winter snow accumulation? This would be an important information.

There are no reliable measurements of these ratios available for the high-altitude sites.

L259 and Fig. 3: Does that 50 km MATCH-ECLAIRE model resolution actually make sense if the cores are only separated by ~100 km? Maybe focus on the BIODIV version and don't show the low resolution runs? The gradient only seems to be visible in the low resolution plots, this is not sufficiently discussed.
L262: Again, what is the benefit of the MATCH-ECLAIRE, if it underestimates the altitudes in such way? Please consider focusing this paragraph only on the high-res version.

The MATCH-ECLAIRE set up is unique in the sense that it provides consistent pan-European trends of atmospheric concentrations and deposition of major atmospheric constituents over the period 1900-2020. Due to computational constraints and uncertainties in the input data during the early period this set-up was operated on relatively coarse resolution.

The MATCH-BIODIV simulations, on the other hand, use more recent input data, available on higher spatial resolution, which enable more detailed analysis of the spatial variation of sulphur- and nitrogen containing species across central Europe, but it is only available from 1987 onwards.

The two datasets when used together allows for a more comprehensive understanding of both spatial and temporal variations. We will improve the discussion on the modelling data sets to reflect these considerations.

The fact that the gradient is only visible in the low resolution model (MATCH-ECLAIRE), but not in the high-res MATCH-BIODIV model is already explained in lines 260-271.

L272: Please reformulate the first sentence. Is this a conclusion from the fact that the west-east gradient was ruled out?

Yes! We will reformulate the sentence accordingly.

L282: Given the close spatial proximity but very different accumulation regimes: how do the CG and GG records compare? This would be important to know in the course of the data evaluation.

As described in 3.2.-3.4. and fig. 2 concentrations agree for CG, GG, and FH. We will add a discussion accordingly.

L284ff: In my opinion this last statement of the paragraph questions the representativity of one single record (at least in total concentration apart from trend) for the region. I think this is contrasting one of the main messages of the paper. This needs further discussion and assessment.

In response to the previous referee comments we will clarify in chapter 3 that representativeness was investigated comparing a) absolute concentration levels and b) long-term concentration trends. Our conclusion is that every ice core from the different sites in the European Alps provides a representative signal of the trend of anthropogenic pollution in Western European countries. This does not hold for the magnitude in concentrations, for which CDD shows generally lower values,

mainly caused by a different seasonality in preserved precipitation. We will adjust this part of the conclusion, as suggested by the referee.

L297f: What about the CG15 peak at about 1920 in Fig. 2? This is also visible in NO3 and NH4, If it still shows up in the 10 year average, I would consider it significant enough to be discussed. Please comment.

The idea of the paper is not to discuss and compare single peaks but to focus on general features like concentration levels and trends. Therefore we do not discuss single 10-year maxima such as the CG15 1910-20 peak in $NH_4^+$, $NO_3^-$, $exSO_4^{2-}$, or the CG95 1880-1890 peak of Pb.

L326ff: Why should these uncertainties only be relevant in the late 20th century? There is almost a factor of 2 difference between observation and model in the modern section. This needs more critical discussion.

There are no differences between model and ice-core trends until end of 20$^{th}$ century. Potential factors for the deviation after this time are explained already (potential uncertainties in NOx emissions estimates or model misrepresentation of the $HNO_3$/$NO_3^-$ partitioning between gas and particle phase and of the atmospheric chemistry).

L339f: if analytical uncertainty is the reason, this is not sufficiently reflected in the uncertainty envelopes. Again, there are excursions for CG15 around 1910-1920 and FH ~1860. This needs a better assessment than the speculative comments. Please extend.

We agree that "analytical uncertainty" is not the correct term and will remove it from the discussion. What we referred to is a potential contamination from the laboratory air, which is not included in the uncertainty envelope. We will clarify that in the revised version.

L400: if the model emission reductions were too optimistic, could this be corrected and re-evaluated to find a better agreement? Please comments

The idea of the paper was to compare ice-core data with available model data. The re-evaluation of model data to fit the ice-core data is out of the scope of this paper and will be tackled in future studies.

L408: Typo: Methods of analysis

Will be changed

L408: you could add "to the CG cores" after "similar trend"

Will be added.

L408f: please clarify what you mean by "elevated Pb levels in the earlier…" does this refer to the relative change? the total concentration is lower anyway. The same applies for Cd.

Yes, it refers to the relative change, we will correct this for Pb and Cd.

L428f: If these are the locations of the smelters, why is there no larger impact in the CDD cores visible? Please comment

The distance of the French smelters to the CDD site is comparable to the distance of the Swiss smelters to the other three sites.

L442f: There is no sufficient answer given to the question of representativity throughout the manuscript. This needs more specific discussion.
L449f: This representativity is in the current stage of the study only true for the long-term trends, but not for the absolute concentrations. This needs clarification.
L455ff: Does this mean that CDD is less representative for the region than the other cores? This also needs more critical assessment and discussion.

See answers to general comments and to line284.

L458-462: This paragraph also needs some evaluation .It only lists the different TOE. What are the consequences for the present day situation?

The present day situation is discussed in the following paragraph (L464-471)

L476-L479: Please reformulate this final statement. It is common knowledge that beyond the instrumental era ice cores are the main tool to reconstruct aerosol deposition. What especially is the knowledge gain emerging from this study? Is it representativity of the single records? Or rather pinning down the onsets of pollution? This needs clarification (see also general comment).

In our study we show for the first time, how consistent the Western European air pollution trends are recorded in single Alpine ice cores. We were able to refine the history for different compounds by pinning down their onset of pollution, maxima, and recent changes. The fact that only four of seven pollutant records do agree with available model estimates illustrates the necessity to include such ice core-based reconstructions for model constrain. We will adapt the final statement accordingly.

Table 1: It would help to add a column with the core length and / or estimated maximum age at the bottom.

Since we use only a part of the respective ice cores covering the period 1750-2015, we think the total core length as well as the maximum age at the bottom is not relevant. This information is available in the given references.

**References:**

Bohleber, P., Wagenbach, D., Schöner, W., & Böhm, R. (2013). To what extent do water isotope records from low accumulation Alpine ice cores reproduce instrumental temperature series? Tellus B: Chemical and Physical Meteorology, 65.

Eckhardt, S., Cassiani, M., Evangeliou, N., Sollum, E., Pisso, I., and Stohl, A.: Source-receptor matrix calculation for deposited mass with the Lagrangian particle dispersion model FLEXPART v10.2 in backward mode, Geosci. Model Dev., 10, 4605-4618, 10.5194/gmd-10-4605-2017, 2017.

Engardt, M., Simpson, D., Schwikowski, M., and Granat, L.: Deposition of sulphur and nitrogen in Europe 1900-2050. Model calculations and comparison to historical observations, Tellus B, 69, 10.1080/16000889.2017.1328945, 2017.

Preunkert, S., Wagenbach, D., and Legrand, M.: A seasonally resolved alpine ice core record of nitrate: Comparison with anthropogenic inventories and estimation of preindustrial emissions of NO in Europe, J. Geophys. Res.-Atmos., 108, 4681, 2003.

Schwikowski, M., Döscher, A., Gäggeler, H. W., and Schotterer, U.: Anthropogenic versus natural sources of atmospheric sulphate from an Alpine ice core, Tellus B, 51, 938-951, 10.3402/tellusb.v51i5.16506, 1999b.

Seinfeld, J. H. and Pandis, S. N.: Atmospheric Chemistry and Physics: From Air Pollution to Climate Change, John Wiley & Sons, Hoboken 2016.

Sierra Hernandez, M.R.,  P. Gabrielli, E. Beaudon, A. Wegner, L. G. Thompson, Atmospheric depositions of natural and anthropogenic trace elements on the Guliya ice cap (northwestern Tibetan Plateau) during the last 340 years. Atmos. Environ. 176, 91–102 (2018).

---

## Author Comment (AC3)

The authors present a synthesis of seven anthropogenic pollutant records from four well-studied ice core sites in the European Alps. The datasets have all been well-vetted in previous publications and the timescales are accurate. If indeed the authors attempt here is the first to compare these datasets, then such a comparison is long overdue and most welcome. Each of the seven chemical constituents show a high degree of correlation among the four sites, as would be expected based on the presented back trajectory and emissions modeling results. The concentration discrepancy at Col de Dome is explained by a higher degree of winter snow preservation, which seems plausible based on the arguments presented. In general, the paper is well written and concise; I don't see any reason not to accept it in its present form. My only possible criticism is that there is nothing particularly new or novel here; but then again, the high degree of agreement between the four sites is powerful and tells a very simple and compelling story.

We really appreciate the positive evaluation of our manuscript by the referee.

In our study we show for the first time, how consistent the Western European air pollution trends are recorded in single Alpine ice cores. In response to both referee comments, we will clarify throughout the manuscript that the main object of the study is the investigation of representativeness, which is novel.

---

## Author Response (AR1)

**1) Reply to editor comment**

Dear Authors

I thank the reviewers for their thorough evaluations and suggestions. I can see in your response that the suggested amendments will make considerable improvements to the paper. Please prepare your revised manuscript, incorporating the changes presented in your response.

The discussion raised by reviewer one about averages is very insightful. I appreciate the extra figures and evaluation to assess the impact of the different averaging approaches. Relating to this, I think a more rigorous approach could be applied to the method of producing annual averages. This is also raised as a separate issue by reviewer 1.

I remain a little confused with the decision to apply a different method of averaging for the CDD site. The explanation from section 2.1.4 is that depositional changes upstream of the site have impacted the winter snow, and that prior to 1890 no winter snow was preserved. However, in section 2.3 you state that for CDD "annual averages we calculated as arithmetic mean of winter and summer concentrations". This implies that it was possible to confidently separate out the summer and winter component and make an average of just those two seasons.

Is this the correct interpretation? Was the data from other times in the year not included in the annual average calculation?

My concern is that this approach is that rather than compensating for changes in deposition (or preservation) it introduces artificial errors into your calculations. For example, what about the years when you state that no winter snowfall was preserved? Are those years just a summer average?

I would recommend that for consistency and clarity you re-calculate the annual averages for CDD in the same way as the other sites. If you feel it is appropriate, you could include the data calculated using both approaches to demonstrate the differences. This would then still require an explanation about the potential loss of winter snow and the impact this might have on the data.

Kind regards
Liz

Dear Liz, we really appreciate your valuable comments and time to review and edit the manuscript.

Based on the evidence given in the previous CDD papers, winter layers at the CDD site were found to generally thin with depth relative to summer layers due to changes in depositional processes upstream of the drilling site (Preunkert et al., 2000). Therefore, annual values were defined as arithmetic mean of winter and summer concentrations in all previous CDD papers and in the first version of our manuscript. Your "interpretation" of the definition of winter and summer layers is not fully correct, in the sum, both layers add up to a total annual layer, with no season missing. The boundaries of the winter half year snow pack have been identified by requiring at least 3 consecutive samples to significantly exceed the 10 ppb NH4+ level. This assignment and its underling reasoning is detailed in Preunkert et al. (2000). For years with missing winter precipitation, interpolated values were used. The time period before 1890 with no winter snow preservation was not considered in our study.

Since the calculation of winter/summer averages was only possible for the CDD site, we agree with your suggestion to consistently form annual averages from total annual layers for all ice cores and discuss the differences between the two averaging procedures in the new supplementary section S2

and Figure S2. We found that for the 10-year means considered in this study, CDD long-term concentration trends for all investigated species agree within the uncertainty envelopes between the two averaging procedures (Figure S2). However, absolute concentrations are factor 1.3-1.5 higher for the total annual layer means compared to the winter/summer averages. As mentioned above, since the calculation of winter/summer averages was only possible for the CDD site, the comparison of concentration records between all sites is now consistently presented in this study based on the averages of total annual layers.

**2) Reply to comments of referee #1**

We would like to thank referee#1 for the willingness to review our manuscript. We really appreciate the detailed comments and suggestions of the referee. They helped to improve the clarity of the manuscript. Please find below our responses to the comments (in blue) and our changes to the revised manuscript (in blue, but italic). Given line numbers refer to the file Eichler et al 2022 rev changes marked.

General comments

In this study the authors present a synthesis of air pollution data from four different High Alpine European ice cores for the last ~250 years. They analyse how representative the single core data are for the common pollution history of the area. They create composite records of 10-year averages for the respective pollution species and calculate "times of emergence" (TOE) when the actual anthropogenic pollution becomes significant. This is an interesting topic and especially the investigation of the representativity of one ice core record for the atmospheric signal of an entire region is very important and was not sufficiently discussed so far. Also the findings of the different times of emergence of the pollution signals in Europe and the present day conditions, including partly failed abatement measurements, are highly relevant and should be published.

The paper is overall well written. The paragraphs follow a clear structure and are easy to follow. However, the general focus of the study remains a little unclear to me. The title and the statements at the end of the discussion (section 3) paragraphs rather suggest that the investigations of the TOE and the evaluation of the present day situation were the main objectives. On the other hand, the authors also dedicate a good deal of the study to the evaluation of the representativity of a single core record for the whole region. In my opinion, both discussions lack depth and critical evaluation. The paper would definitely benefit from a clarification of the main objective and a more focused approach (methodological or interpretation of pollution history). Especially the discussion section (3.2-3.4) is sometimes rather vague and needs some revision. I think the discrepancies between the single core records (Fig. 2) and the resulting consequences for the composite record (and the following interpretation) are not discussed and assessed sufficiently (see specific comments).

The main object of the study is to investigate, how representative one Alpine ice core is to document the pollution history of Western Europe. As suggested by the referee we clarified this throughout the manuscript, where not done already (end of introduction, first chapter of section 3., conclusion). We think that the chosen title does not contradict the main object of the study and prefer to keep it. In chapter 3 (lines 263-265) we clarified that for investigating the representativeness a) concentration levels and b) longer-term concentration variations of BC, Cd, $F^-$, $NH_4^+$, $NO_3^-$, Pb, and

exSO$_4^{2-}$ from the four sites (CDD, CG, FH, and GG) were compared. For a more quantitative and less "vague" comparison of the long-term trends we added a correlation analyses (new Figure 4).

[Figure]

Figure 4: Correlation matrices showing the correlation coefficients r between 10-year concentration records from different ice cores and the Alpine composite (Alp) for the individual pollutants exSO$_4^{2-}$, NO$_3^-$, NH$_4^+$, BC, Cd, Pb, and F$^-$.

According to the comments of the referee we did rearrange the structure of the Results and Discussion part:
- 3. Results and Discussion
general paragraph:
*Average emission sensitivities during the period 2000-2009 derived with FLEXPART, indicate that in general the four ice-core sites CDD, CG, FH, and GG are most sensitive to emissions from the countries surrounding the Alps, i.e. Switzerland, France, Italy, Germany, Austria, Slovenia, and Spain, without substantial discrepancies between the four sites (Figure 1). A noticeable extension of the pollution source area into Spain is recognized for the westernmost CDD, and into larger parts of Germany for the northernmost FH site. To investigate, how representative ice-core records from the four sites are to document the pollution history in the identified source area we systematically compared a) concentration levels (section 3.1.) and b) longer-term variations of of BC, Cd, F$^-$, NH$_4^+$, NO$_3^-$, Pb, and exSO$_4^{2-}$ concentrations (section 3.2.)."*

- 3.1. Magnitudes of concentration levels
  (former chapter 3.1.2)
- 3.2. Long-term variations
general paragraph:
*"Generally, an excellent agreement in the 10-year ice-core longer-term concentration changes (trends) of all investigated pollutants is observed for the four sites CDD, CG, FH, and GG (Figure 2). This is confirmed by correlation analyses, revealing high correlation coefficients (0.84 < r < 1.00) between the respective pollutant records from the different ice cores (Figure 4). All correlations are significant at the 0.001 level. The observed trends are characteristic for the individual seven pollutants, reflecting differences in their respective emission sources, emission histories, the atmospheric chemistry of precursor species, and compound specific implementation and efficiency of*

*emission abatement measures. This is detailed in the following sections (3.2.1-3.2.3), including discussion of how representative the Western European pollution history is captured."*

- chapters 3.2.1 – 3.2.3.
former chapters 3.2.-3.4.
A discussion about potential different features in the records and representativeness of the single records including the results of the correlation analyses was added at the end of each chapter.

My other major point of criticism is the methodological approach for averaging in section 2.3. It is not justified why the authors calculate 10-year averages for the single ice core records. Is that indeed necessary to "average out inter-annual-fluctuations"? (L203). If only looking at a 250 year time range this seems to smooth out a lot of variation. Could another averaging window also be sufficient and maybe provide more detailed insights? This needs some statistical evaluation. The reasoning behind the 10-year averaging seems random, remains unclear so far and will have consequences for the following discussion of TOE and pollution records. This section needs to be extended, clarified and methodologically justified.

We agree with the referee that this point of the chosen averaging window was not explained properly and extended the discussion in section 2.3 and the new supplementary section S1 and Figure S1. There are two main reasons for the chosen 10-year window: (1) averaging out inter-annual fluctuations mainly related to variations in pollutants vertical transport to the high altitudes controlled by atmospheric stability and (2) accounting for the low data resolution and high dating uncertainty in the lower ice core parts (e.g. 10-25 years for the FH in the period 1750-1800).

To select an adequate averaging period, we investigated the influence of the different averaging windows on the CG15 $exSO_4^{2-}$ record in the period 1920-2015 (Fig. S1). Figure S1a illustrates that annual $exSO_4^{2-}$ maxima/minima are in part synchronous with $\delta^{18}O$ maxima/minima. Since $\delta^{18}O$ is a temperature proxy (Bohleber et al., 2013), this reflects a stronger/weaker vertical transport of pollutants from their source regions at lower elevations to the high-altitudes sites during warmer/colder summers. This is still partially visible in the 5-year means (Fig. S1b), e.g. the temperature ($\delta^{18}O$) maximum in the period 1970-75 lead to an amplification of the $exSO_4^{2-}$ concentration at CG compared to what is expected based on the $SO_2$ emission history (Fig. S1e). This modulation of the ice-core record by year-to-year variability of vertical transport/atmospheric stability also produces a larger variability compared to the modelled concentrations for the annual and 5-year means (Fig. S1 d,e), but not for the 10-year averages. The MATCH-ECLAIRE model does not have sufficient resolution to resolve vertical transport.

Thus, we argue that the ice-core 10-year averages (1) are representative for changing pollutant emissions, averaging out the inter-annual (short-term) fluctuations related to temperature dependent pollutant vertical transport to the high-alpine sites and (2) take into consideration the low data resolution and higher dating uncertainty in the pre-industrial period.

[Figure]

Fig. S1: Comparison of CG15 exSO$_4^{2-}$ concentrations (pink) with CG15 $\delta^{18}$O records (panels a-c, black) and with modelled exSO$_4^{2-}$ concentrations (MATCH-ECLAIRE) (panels d-f, turquis) in the period 1920-2015 using different averaging periods (a,d – 1-year, b,e – 5 year, c,f – 10 year).

I have some specific comments, especially about some figures and the respective discussions.

Specific comments

L21: please put "long term" in perspective. For most ice core studies, 250 years would not be considered long term

We put the "long-term" already in perspective by adding the time period (AD 1750-2015), but changed now the wording to "longer-term" throughout the whole manuscript.

L21: please specify "several" ice cores

We specified: *"Depending on the data availability for the different air pollutants, up to five ice cores from four high-Alpine sites located in the European Alps analysed by different laboratories were considered."* (lines 22-24)

L37f: This statement lacks evaluation. What are the consequences from only four core records corresponding well with the modelled trends? Please elaborate

We added potential reasons: *"Only four ice-core composite records (BC, F⁻, Pb, exSO$_4^{2-}$) of the seven investigated pollutants correspond well with modelled trends, suggesting inaccuracies of the*

*emission estimates or an incomplete representation of chemical reaction mechanisms in the models for the other pollutants."* (lines 38-40)

L51: change "ice cores" to "ice core records"

We changed this accordingly (lines 53-54).

L66ff: Maybe a problem of wording: "… indicates common source regions for the different Arctic sites…" I think this is a rather generalized and bold statement, especially because in the next sentence you state that this is not true for e.g. Svalbard. Please clarify.

Was changed to *"This finding indicates common source regions for the different Greenland sites, located in North America and Europe…"* (lines 68-69)

L79f: Reference for huge Himalayan dust deposition?

A reference was added accordingly (Sierra-Hernandez et al., 2018) (line 83).

L87: you could add: Preunkert, S., McConnell, J. R., Hoffmann, H., Legrand, M., Wilson, A. I., Eckhardt, S., et al. (2019). Lead and antimony in basal ice from Col du Dome (French Alps) dated with radiocarbon: A record of pollution during antiquity. Geophysical Research Letters, 46, 4953– 4961. https://doi.org/10.1029/2019GL082641

The suggested reference was added (line 90).

L90f: The sentence about the downward trend is somewhat repetitive throughout the manuscript. Please shorten.

The sentence was shortened (lines 92-93).

L93: Please specify that "annual" concentrations at Colle Gnifetti are also heavily biased towards the summer season because of major wind erosion of winter snow.

This point is already detailed in the chapter 2.1.1.

L102: which were the "different laboratories"?

This is different for specific pollutants and is already mentioned in chapter 2.2.

L114: Can you add a reference for the atmospheric lifetime of the species? Are they the same for all different pollutants?

Atmospheric lifetimes vary depending on the conditions (occurrence of precipitation). We think for the purpose of this study, an average lifetime in the order of a few days for all the investigated species is a reasonable statement. We added a reference accordingly (Seinfeld and Pandis, 2016). (line 116)

L137: please add: "… parts or all of the winter snow cover…"

Was added (line 139).

L138: add per year after w.eq.

"Annual" is already mentioned in this sentence ("annual net accumulation rates of 0.2-0.5 m water equivalent") (line 140).

L139: Please weaken this statement to "… ice at bedrock can be more than…" and cite additionally: Hoffmann, H., Preunkert, S., Legrand, M., Leinfelder, D., Bohleber, P., Friedrich, R., & Wagenbach, D. (2018). A New Sample Preparation System for Micro-14C Dating of Glacier Ice with a First Application to a High Alpine Ice Core from Colle Gnifetti (Switzerland). Radiocarbon, 60(2), 517-533. doi:10.1017/RDC.2017.99

Statement was weakened (*"and the observation that older than 15000-year old ice was found at the bedrock."*) and the suggested reference added. (line 141)

L173: were the IC measurements discrete or quasi-continuous? What was the spatial resolution? Please specify.

"*on discrete samples*" was added (line 178). The spatial resolution was different from core to core and is detailed in the different references given in table 1 and 2.

L174: Is there a reference for the IUP measurements?

The given reference (Preunkert et al., 2001a) does also include the IUP measurements.

Section 2.2.3: Are there no measurements available for the other cores? Please explain.

For this study we used records that were published before end of 2021. A comment was added accordingly (lines 202-203).

L197f: Is this sea salt and mineral dust correction valid for such rather continental sites?

Yes it is based on the evidence in the cited paper (Schwikowski et al., 1999b).

L201: It is unclear what happened after the separate calculation of summer and winter mean concentrations. Were they also evaluated separately? Please extend and clarify.

Based on the evidence given in the previous CDD papers, winter layers at the CDD site were found to generally thin with depth relative to summer layers due to changes in depositional processes upstream of the drilling site (Preunkert et al., 2000). Therefore, annual values were defined as arithmetic mean of winter and summer concentrations in all previous CDD papers and in the first version of our manuscript. Since the calculation of winter/summer averages was only possible for the CDD site, we agree with the editor's suggestion (see above) to consistently form annual averages from total annual layers for all ice cores and discuss the differences between the two averaging procedures in the section S2 and Figure S2. We found that for the 10-year means considered in this study, CDD long-term concentration trends for all investigated species agree within the uncertainty envelopes between the two averaging procedures (Figure S2). However, absolute concentrations are factor 1.3-1.5 higher for the total annual layer means compared to the winter/summer averages. As mentioned above, since the calculation of winter/summer averages was only possible for the CDD site, the comparison of concentration records between all sites is consistently presented in this study based on the averages of total annual layers.

L203: See general comments. How were the 10-year averages calculated? Just arithmetic mean? Weighted? Please extend and explain.

Arithmetic means were calculated, a comment was added (line 208).

L205: see general comments "should better reflect" is no sufficient justification for the 10-year averaging. Please assess better and explain reasoning.

See detailed answer to this general comment above. We added a new supplementary section S1 to justify the choice of 10-year averages.

L210: The reason for not including the GG records does not convince me. 50 years out of 250 is already 20%. If this is considered too short, the GG core should have been left out completely.

For calculating z-scores, averages and stdev of a common overlapping period have to be considered. We consider the 50 years period covered by the GG core too short (just five data points) to obtain representative values of these two measures. Following the referees comment for L282, we prefer to leave the GG records in Fig. 2 and related discussion in the manuscript to compare concentration data from this high-accumulation site (~2.7 m weq/yr) with the ones from the close-by low-accumulation CG site (~0.2-0.5 m weq/yr).

L211: Again, are there no BC /EC records available for the other cores? Clarify.

There are no other published continuous records.

L219 and Fig. 1: It is really hard to see significant differences between the FLEXPART results for the different locations. Maybe this figure could benefit from a larger zoom on the locations and / or a different colour scheme if the goal is to show the very small large-scale variations?

There are no substantial differences in the pollutant source areas for the different sites (line 260). Since the goal here is to show common general source areas in Western Europe, we prefer to leave Fig 1 as it is.

L221: "Most species" – which ones are not present as sub-micrometer aerosols? Specify.

We changed "*most species*" to "*all species are predominantly present...*" (line 237).

L229: how exactly was this calculated? Please elaborate.

A detailed description of the calculation of the source-receptor relationship for deposited mass in FLEXPART is provided by Eckhardt et al. (2017). We added a corresponding sentence (lines 235-236).

L234: spatial distribution referring to what region? Please specify

The MATCH model was applied to a European domain for both datasets (MATCH-BIODIV and MATCH-ECLAIRE), as indicated in the opening sentence of section 2.5. We used these simulations but focusing on the spatial variation in the Alps in this study (e.g. Fig 3). We clarified that the modelled spatial distribution of $exSO_4^{2-}$ concentration in the Alps, used in the present study, was extracted from these simulations (line 251).

L238: which climate simulations? The whole paragraph 2.5 is rather vague. Please specify.

The climate simulations (and emission data) used for the MATCH-ECLAIRE simulations are described in the Engardt et al. (2017) paper, while the MATCH-BIODIV will be described in a forthcoming paper and in other present work. It is outside the scope of the present paper to explain all details of the atmospheric chemistry simulations here, but we updated paragraph 2.5 with more details (lines 246-257).

L243: how do you define long-term-trend? What would happen for a smaller / larger averaging window?

We changed this term to "longer-term variations" (line 265, 309). The justification for the choice of the 10-year averages is given above and in the new supplementary section S1.

L254f: Please reformulate this sentence. It seems odd if you write "could be explained" when you dismiss this possibility later in the text. Also the west-east gradient seems confusing in this context, when in Fig. 1 you mention that for CDD the source region is extended to the west.

We reformulated the sentence: "*Potential reasons for the lower CDD concentrations are…*" (line 268)

L255: Do you have estimates for the ratio of the summer / winter snow accumulation? This would be an important information.

There are no reliable measurements of these ratios available for the high-altitude sites except CDD.

L259 and Fig. 3: Does that 50 km MATCH-ECLAIRE model resolution actually make sense if the cores are only separated by ~100 km? Maybe focus on the BIODIV version and don't show the low resolution runs? The gradient only seems to be visible in the low resolution plots, this is not sufficiently discussed.
L262: Again, what is the benefit of the MATCH-ECLAIRE, if it underestimates the altitudes in such way? Please consider focusing this paragraph only on the high-res version.

The MATCH-ECLAIRE set up is unique in the sense that it provides consistent pan-European trends of atmospheric concentrations and deposition of major atmospheric constituents over the period 1900-2020. Due to computational constraints and uncertainties in the input data during the early period of this set-up was operated on relatively coarse resolution.

The MATCH-BIODIV simulations, on the other hand, use more recent input data, available on higher spatial resolution, which enable more detailed analysis of the spatial variation of sulphur- and nitrogen containing species across central Europe, but it is only available from 1987 onwards.

The two datasets when used together allows for a more comprehensive understanding of both spatial and temporal variations. We adapted paragraph 2.5. accordingly. We prefer to leave the results for the MATCH-ECLAIRE model in this section, since modeling of the temporal trends 1900-2020 was only possible based on this model.

The fact that the gradient is only visible in the low resolution model (MATCH-ECLAIRE), but not in the high-res MATCH-BIODIV model is already explained in lines 270-283.

L272: Please reformulate the first sentence. Is this a conclusion from the fact that the west-east gradient was ruled out?

Yes! We reformulated the sentence accordingly (line 287) and added bullets (1) and (2) to better differentiate between the two reasons (lines 268-269, 287).

L282: Given the close spatial proximity but very different accumulation regimes: how do the CG and GG records compare? This would be important to know in the course of the data evaluation.

As described in 3.2.-3.4. and Fig. 2 concentrations agree for CG, GG, and FH. We added a discussion accordingly (lines 303-306).

L284ff: In my opinion this last statement of the paragraph questions the representativity of one single record (at least in total concentration apart from trend) for the region. I think this is contrasting one of the main messages of the paper. This needs further discussion and assessment.

In response to the previous referee comments we clarified in chapter 3 (lines 263-265) that representativeness was investigated comparing a) concentration levels and b) longer-term concentration variations. Our conclusion is that every ice core from the different sites in the European Alps provides a representative signal of the trend of anthropogenic pollution in Western European countries. This does not hold for the magnitude in concentrations, for which CDD shows generally lower values, mainly caused by a different seasonality in preserved precipitation. We adapted the abstract and conclusion part accordingly.

L297f: What about the CG15 peak at about 1920 in Fig. 2? This is also visible in NO3 and NH4, If it still shows up in the 10 year average, I would consider it significant enough to be discussed. Please comment.

The idea of the paper is not to discuss and compare single peaks but to focus on general features like concentration levels and trends. Therefore we do not discuss single 10-year maxima such as the CG15 1910-20 peak in $NH_4^+$, $NO_3^-$, $exSO_4^{2-}$, or the CG95 1880-1890 peak of Pb.

L326ff: Why should these uncertainties only be relevant in the late 20th century? There is almost a factor of 2 difference between observation and model in the modern section. This needs more critical discussion.

There are no differences between model and ice-core trends until end of 20[th] century. Potential factors for the deviation after this time are explained already (potential uncertainties in NOx emissions estimates or model misrepresentation of the $HNO_3/NO_3^-$ partitioning between gas and particle phase and of the atmospheric chemistry).

L339f: if analytical uncertainty is the reason, this is not sufficiently reflected in the uncertainty envelopes. Again, there are excursions for CG15 around 1910-1920 and FH ~1860. This needs a better assessment than the speculative comments. Please extend.

We agree that "analytical uncertainty" is not the correct term and remove it from the discussion. What we referred to is a potential contamination from the laboratory air, which is not included in the uncertainty envelope. We clarify this (line 368-369).

L400: if the model emission reductions were too optimistic, could this be corrected and re-evaluated to find a better agreement? Please comments

The idea of the paper was to compare ice-core data with available model data. The re-evaluation of model data to fit the ice-core data is out of the scope of this paper and will be tackled in future studies.

L408: Typo: Methods of analysis

Was changed accordingly (line 445).

L408: you could add "to the CG cores" after "similar trend"

Was changed accordingly (line 445).

L408f: please clarify what you mean by "elevated Pb levels in the earlier…" does this refer to the relative change? the total concentration is lower anyway. The same applies for Cd.

Yes, it refers to the relative change, we corrected this (line 445).

L428f: If these are the locations of the smelters, why is there no larger impact in the CDD cores visible? Please comment

The distance of the French smelters to the CDD site is comparable to the distance of the Swiss smelters to the other three sites.

L442f: There is no sufficient answer given to the question of representativity throughout the manuscript. This needs more specific discussion.
L449f: This representativity is in the current stage of the study only true for the long-term trends, but not for the absolute concentrations. This needs clarification.
L455ff: Does this mean that CDD is less representative for the region than the other cores? This also needs more critical assessment and discussion.

See answers to general comments and to L284.

L458-462: This paragraph also needs some evaluation .It only lists the different TOE. What are the consequences for the present day situation?

The present day situation is discussed in the following paragraph (L507-515)

L476-L479: Please reformulate this final statement. It is common knowledge that beyond the instrumental era ice cores are the main tool to reconstruct aerosol deposition. What especially is the knowledge gain emerging from this study? Is it representativity of the single records? Or rather pinning down the onsets of pollution? This needs clarification (see also general comment).

We find that longer-term concentration variations of seven investigated air pollutants feature a uniform timing in species-dependent anthropogenic impact at four sites in the European Alps. Our results demonstrate that all ice-core records provide a representative signal of anthropogenic pollution changes in Western European countries. Thus, in our study we show for the first time, how consistent this pollution history is recorded in different Alpine ice cores. We were able to refine the

history for different compounds by pinning down their onset of pollution, maxima, and recent changes. We adapted the summary and conclusion accordingly.

Table 1: It would help to add a column with the core length and / or estimated maximum age at the bottom.

Since we use only a part of the respective ice cores covering the period 1750-2015, we think the total core length as well as the maximum age at the bottom is not relevant. This information is available in the given references.

**3) Reply to comments of referee #2**

The authors present a synthesis of seven anthropogenic pollutant records from four well-studied ice core sites in the European Alps. The datasets have all been well-vetted in previous publications and the timescales are accurate. If indeed the authors attempt here is the first to compare these datasets, then such a comparison is long overdue and most welcome. Each of the seven chemical constituents show a high degree of correlation among the four sites, as would be expected based on the presented back trajectory and emissions modeling results. The concentration discrepancy at Col de Dome is explained by a higher degree of winter snow preservation, which seems plausible based on the arguments presented. In general, the paper is well written and concise; I don't see any reason not to accept it in its present form. My only possible criticism is that there is nothing particularly new or novel here; but then again, the high degree of agreement between the four sites is powerful and tells a very simple and compelling story.

We really appreciate the positive evaluation of our manuscript by the referee.

We find that longer-term concentration variations of seven investigated air pollutants feature a uniform timing in species-dependent anthropogenic impact at four sites in the European Alps. Our results demonstrate that all ice-core records provide a representative signal of anthropogenic pollution changes in Western European countries. Thus, in our study we show for the first time, how consistent this pollution history is recorded in different Alpine ice cores. We were able to refine the history for different compounds by pinning down their onset of pollution, maxima, and recent changes.

In response to both referee comments, we clarified throughout the manuscript that the main object of the study is the investigation of representativeness, which is novel.

**4) Reply to comments of Michael Sigl**

This study combines and discusses air pollution records reconstructed from four different ice-core sites in the European Alps. My comment suggests some technical clarifications concerning the ice-core records from Colle Gnifetti which I invite the authors to address in the revision.

Thanks Michael for the suggested technical clarifications. We added them to the manuscript accordingly.

L 142: Please add as citation for the dating Sigl et al., (2018) providing necessery details about the individual ice cores (Table 1) and the final dating (Supplementary) used to derive the NO3, NH4, rBC

and SO4 records used for this study together with access to the underlying high-resolution datasets (https://doi.org/10.1594/PANGAEA.894787).

The reference Sigl et al. (2018) is already included in Tables 1 and 2, but we also added it to line 145.

L 172: Are these new IC analyses for the CG15 ice core extending back to 1750; or are these the upper 12m of CG15 merged with the records from the two CG03 ice cores (CG03A and CG03B)? Please clarify; see Table 1 in Sigl et al., (2018) for details. If these are new analyses, how was the dating of CG15 done, and how do the records compare relative to the CG03 records during the period of overlap? If the latter, which records did you use for your analyses? CG03A, CG03B or a stacked record of both?

These are the records from the upper 12m of CG15 merged with those from the CG03A core. Details were added to Table 2.

L 187: Same question here: Are these new rBC analyses from the CG15 ice core?  The reference to Sigl et al., (2018) suggests these are from CG03B.

These are the records from the upper 12m of CG15 merged with those from the CG03B core. Details were added to Table 2.

L 196: This now answers the two questions noted above. To avoid misunderstanding, I suggest to correct or clarify the meaning of the core labels in the sections above and in Table 1 and 2, since the CG15 ice core is actually encompassing the time period 1750-2015 overlapping with CG03 ice cores.

Details were added to Table 2.

L 487: Add "Data from CG ice cores are available at PANGAEA, doi.org/10.1594/PANGAEA.894787" to acknowledge the work generating these measurements.

We added this sentence to the final version (line 529-530).

Table 2: Suggest to replace "CG15" with "CC03_CG15" or similar, to show this is the CG03 ice core updated by the CG15 ice core.

We replaced CG15 with "CG03A+CG15" for $NH_4^+$, $NO_3^-$, $SO_4^{2-}$ and with "CG03B+CG15" for EC, rBC in Table 2.

**References:**

Bohleber, P., Wagenbach, D., Schöner, W., & Böhm, R. (2013). To what extent do water isotope records from low accumulation Alpine ice cores reproduce instrumental temperature series? Tellus B: Chemical and Physical Meteorology, 65.

Eckhardt, S., Cassiani, M., Evangeliou, N., Sollum, E., Pisso, I., and Stohl, A.: Source-receptor matrix calculation for deposited mass with the Lagrangian particle dispersion model FLEXPART v10.2 in backward mode, Geosci. Model Dev., 10, 4605-4618, 10.5194/gmd-10-4605-2017, 2017.

Engardt, M., Simpson, D., Schwikowski, M., and Granat, L.: Deposition of sulphur and nitrogen in Europe 1900-2050. Model calculations and comparison to historical observations, Tellus B, 69, 10.1080/16000889.2017.1328945, 2017.

Preunkert, S., Wagenbach, D., Legrand, M., and Vincent, C.: Col du Dome (Mt Blanc Massif, French Alps) suitability for ice-core studies in relation with past atmospheric chemistry over Europe, Tellus B, 51, 993-1012, 2000.

Preunkert, S., Wagenbach, D., and Legrand, M.: A seasonally resolved alpine ice core record of nitrate: Comparison with anthropogenic inventories and estimation of preindustrial emissions of NO in Europe, J. Geophys. Res.-Atmos., 108, 4681, 2003.

Schwikowski, M., Döscher, A., Gäggeler, H. W., and Schotterer, U.: Anthropogenic versus natural sources of atmospheric sulphate from an Alpine ice core, Tellus B, 51, 938-951, 10.3402/tellusb.v51i5.16506, 1999b.

Seinfeld, J. H. and Pandis, S. N.: Atmospheric Chemistry and Physics: From Air Pollution to Climate Change, John Wiley & Sons, Hoboken 2016.

Sierra Hernandez, M.R., P. Gabrielli, E. Beaudon, A. Wegner, L. G. Thompson, Atmospheric depositions of natural and anthropogenic trace elements on the Guliya ice cap (northwestern Tibetan Plateau) during the last 340 years. Atmos. Environ. 176, 91–102 (2018).

Sigl, M., Abram, N. J., Gabrieli, J., Jenk, T. M., Osmont, D., and Schwikowski, M.: 19th century glacier retreat in the Alps preceded the emergence of industrial black carbon deposition on high-alpine glaciers, Cryosphere, 12, 3311-3331, 10.5194/tc-12-3311-2018, 2018.

---

## Author Response (AR2)

Dear Authors

Thank you for making the requested changes to the manuscript. I feel it is much improved and am happy to accept it for publication.
I appreciate the additional explanation for the annual averages. The previous method may be suitable for a publication focused on a single site, however, to maintain your objective of presenting a consistent history across sites it is important that the method is also consistent. Now that you have applied a uniform averaging method, I feel more confident that this objective has been met.

Please make the suggested amendments and upload a revised version of the manuscript.

Kind regards
Liz

Dear Liz, thanks again for your valuable comments and the time to review and edit our manuscript! We implemented all your recommended changes into the new version of the manuscript. Line numbers refer to the manuscript version with marked changes.

I have some minor recommendations and corrections.
In your explanation of the winter to summer ratio, I don't think this number should be 1. A ratio is a comparison of two things, so you must include both numbers for it to be expressed as a ratio. If you mean that summer and winter accumulation is equal, then the ratio is 1:1. Please update this with the two numbers relative to each other.

We added the ratios accordingly (line 174).

In the final version, please check the use of the word "allowing". I'd recommend removing in line 119 and 125. e.g line 119 …changes, to test how ….

We removed "allowing" in line 119 and 125.

Line 170 update "rather regular" to just regular.

We deleted "rather" in line 172.

Line 208 – starting a sentence with a number and needs restructuring – change the use of they showed to be most…

We combined the two sentences: "*Then, based on these annual means, 10-year arithmetic concentration averages were calculated, which are representative of pollutant emissions, smoothing out inter-annual (short-term) fluctuations related to temperature dependent vertical transport to the high-alpine sites (see S1 and Fig. S1).*" (lines 211-213)

Line 212 - remove i.e age.

We removed "i.e. age" (line 214).

Line 307 – replace like with such as? Or in brackets e.g. wind erosion.

We replaced "like" with "such as" (line 314).

Line 394 – rephrase – "…, the concentration of all three major inorganic …."

We rephrased it to "…, the concentrations of all three major inorganic aerosol species…" (line 402) and replaced "major ions" by major inorganic aerosol species" in line 486.

Line 395 & 396 - start sentence with The

We start the sentence now with "The lowest correlation…"

Line 4756 – remove also

We removed "also".

In addition, we added the proper DOI citation to all references.